# A high-resolution atlas of the brain predicts lineage and birth order underlying neuronal identity

## Graphical abstract

## Highlights

- Single-cell atlas reveals 4,167 neuronal subtypes in adult *Drosophila* brain

- Lineage and birth order shape adult neuron transcriptomic identities

- Distinct TF families act along spatial and temporal axes of neuron specification

- Intersectional genetic tools bridge gene expression to anatomy and circuit function

## Authors

Aaron M. Allen, Megan C. Neville,
Tetsuya Nojima, Faredin Alejevski,
Devika Agarwal, David Sims,
Stephen F. Goodwin

## Correspondence

aaron.allen@cncb.ox.ac.uk (A.M.A.),
stephen.goodwin@cncb.ox.ac.uk
(S.F.G.)

## In brief

Using an integrated single-cell transcriptomic atlas of the adult *Drosophila* central brain, Allen et al. uncover over 4,000 transcriptionally distinct neuronal subtypes. The authors show that adult neuronal identity is shaped by a combination of spatial and temporal developmental programs, with distinct transcription factor families acting along each axis. These findings provide a framework for linking gene expression to neural lineage, anatomy, and function.

Allen et al., 2026, Cell Genomics 6, 101103
March 11, 2026 © 2025 The Author(s). Published by Elsevier Inc.

# Cell Genomics

CellPress

Resource

# A high-resolution atlas of the brain predicts lineage and birth order underlying neuronal identity

Aaron M. Allen,[1,3,*] Megan C. Neville,[1,3] Tetsuya Nojima,[1] Faredin Alejevski,[1] Devika Agarwal,[2] David Sims,[2] and Stephen F. Goodwin[1,4,*]
[1]Centre for Neural Circuits and Behaviour, University of Oxford, Oxford, Oxfordshire OX1 3TA, UK
[2]MRC Computational Genomics Analysis and Training Programme (CGAT), MRC Centre for Computational Biology, MRC Weatherall Institute of Molecular Medicine, John Radcliffe Hospital, Oxford, Oxfordshire OX3 9DS, UK
[3]These authors contributed equally
[4]Lead contact
*Correspondence: aaron.allen@cncb.ox.ac.uk (A.M.A.), stephen.goodwin@cncb.ox.ac.uk (S.F.G.)

## SUMMARY

Gene expression shapes the nervous system at every biological level, from molecular and cellular processes defining neuronal identity and function to systems-level wiring and circuit dynamics underlying behavior. Here, we generate the first high-resolution, single-cell transcriptomic atlas of the adult *Drosophila melanogaster* central brain by integrating multiple datasets, achieving an unprecedented 10-fold coverage of every neuron in this complex tissue. We show that a neuron's genetic identity overwhelmingly reflects its developmental origin, preserving a genetic address based on both lineage and birth order. We reveal foundational rules linking neurogenesis to transcriptional identity and provide a framework for systematically defining neuronal types. This atlas provides a powerful resource for mapping the cellular substrates of behavior by integrating annotations of hemilineage, cell types/subtypes, and molecular signatures of underlying physiological properties. It lays the groundwork for a long-sought bridge between developmental processes and the functional circuits that give rise to behavior.

## INTRODUCTION

Unraveling the cellular diversity of the nervous system is crucial for understanding how complex behaviors emerge. Although all cells in the nervous system share the same genetic blueprint, differentiation and specialization are driven by differential transcriptional regulation of the genome.[1,2] In *Drosophila melanogaster*'s central brain, subtle transcriptional variations sculpt intricate, highly specialized neural circuits that support complex cognitive functions and behaviors. Mapping the transcriptomes of all neurons within this tissue is essential to uncovering the specific genes and regulatory networks active in each cell type, providing insights into the molecular mechanisms that underlie information processing and adult behaviors.

The adult *Drosophila* central brain develops from approximately 100 pairs of bilaterally symmetric neuronal stem cells, known as neuroblasts, each producing a highly stereotyped lineage of neurons.[3–7] Neuroblasts express unique combinations of transcription factors (TFs), creating a "genetic address" that helps define cell types.[8,9] During neurogenesis, asymmetric Notch signaling (ON or OFF) assigns distinct fates to sibling neurons, giving rise to hemilineages that define the anatomical subunits within the brain. The central brain contains two main types of neuroblasts, type I and type II, distinguished by their division patterns and the complexity of their resulting neuronal lineages. Type I neuroblasts generate lineages of approximately

200 neurons, while a small subset of type II neuroblasts (eight per hemisphere) produces larger lineages, averaging over 600 neurons.[10,11] Type II lineages, using intermediate neural progenitor cells, are considered analogous to those in the primate cortex, allowing a single neuroblast to produce many more cells.[12,13]

In both types of neuroblast lineages, tightly regulated temporal and spatial gene expression patterns establish neuronal identities.[14,15] Temporal TFs (tTFs) are pivotal in this process, marking distinct stages of neuroblast division and shaping the fate of progeny neurons. While these developmental dynamics are well characterized during embryogenesis and larval stages, their persistence and influence on adult neuronal identity remain less understood. Investigating whether these developmental signatures are retained or modified in the adult brain is essential for understanding the continuity between neurodevelopment and mature brain function.[16]

In this study, we present the most comprehensive single-cell transcriptional atlas of *Drosophila melanogaster* adult central brain neurons to date, generated using single-cell RNA sequencing (scRNA-seq). To sufficiently represent the depth and complexities of cell types within the central brain, we performed a meta-analysis, combining our newly described scRNA-seq data with publicly available data. This analysis generated an atlas of 329,466 neurons, representing an average 9.8× depth of coverage of every single neuron of the central brain.

Our findings reveal that adult central brain neurons are largely transcriptionally defined by their developmental histories. Neuroblast hemilineages emerge as the primary genetic units that define transcriptionally distinct neuronal cell types in the adult central brain. Within these hemilineages, we observe that transcriptional signatures linked to different birth-order windows are maintained into adulthood. Notably, early-born neurons exhibit distinct transcriptional profiles compared to late-born neurons, uncovering a novel axis of neuronal diversity in the adult brain that likely reflects differing developmental demands.

Variation in gene expression drives plasticity, learning, and adaptation, while more drastic gene disruptions contribute to neurological disorders; thus, understanding the underlying genetic mechanisms active in distinct neurons within the brain is essential to linking molecular processes to circuit function. Moreover, leveraging the distinct molecular signatures of neuronal subtypes enables targeted genetic access, facilitating the causal investigation of circuit dynamics. Our high-resolution single-cell transcriptomic atlas of the adult *Drosophila* central brain represents a significant advancement in neurogenomics. By providing an unprecedented level of resolution and integrating genetic, developmental, and anatomical data, this Resource provides a comprehensive framework for dissecting the molecular logic of neuronal diversity. More broadly, our findings offer a platform for comparative studies across species, informing general principles of brain development, organization, and evolution.

## RESULTS

### Generating a high-depth single-cell transcriptomic atlas of the adult central brain

To capture the transcriptional diversity of neuronal cell types within the adult *Drosophila* central brain, we first generated a sexed scRNA-seq dataset, which yielded ~1.6× coverage of central brain neurons (Figure S1; Table S1). Given this tissue's complexity of neuronal cell types, we concluded that significantly greater coverage was required to refine neural identity. We therefore integrated multiple publicly available datasets[17–27] and generated a meta-atlas of >1 million cells/nuclei from the adult *Drosophila* head (Figures S2A and S2B).

We refined this atlas in three steps (Figures S2C–S2G). First, we annotated and removed non-neuronal cells, yielding an ~700,000 neuronal head meta-atlas (Figure S2D). Next, we identified and excluded optic lobe and peripheral neurons (Figures S2E and S2F). The remaining central brain neurons were re-clustered, resulting in a high-resolution transcriptomic atlas of 329,466 central brain neurons, achieving an unprecedented 9.8× average depth of coverage of every neuron in the central brain (Figures 1A, S2G, S3, S4, and S5A–S5C). Alternative integration strategies (see STAR Methods) yielded similar results, supporting the robustness of our findings.

Using established marker genes, we assigned broad cellular identities to 246 transcriptionally distinct neuronal clusters (Figure 1B). The proportional makeup of broad cell types was consistent with previous reports[19,20,28] (Figures 1C and S5A) and includes robust coverage of both sexes, with sex differences explored in detail in a companion study.[29] Notably, central brain neurons from the Fly Cell Atlas (FCA) comprise only

3.8% of the neurons recovered in our dataset.[25] In the FCA dataset, most central brain neurons remained unresolvable and therefore unannotated,[24] indicating that the central brain's complexity required far greater coverage. Indeed, no individual dataset used here was sufficient to describe this complexity. Instead, our cost-effective approach of integrating multiple datasets achieved the necessary resolution to identify and annotate the vast majority of central brain neurons. To estimate our atlas's cellular depth of coverage, we compared our annotated neuronal cell counts with anatomical estimates from the FlyWire connectome.[10,11] These ranged from scarce populations of four neurons per brain (e.g., Hug-RG or ExR1) to all ~34,000 neurons of the central brain and achieved a depth of coverage of 9.8× with perfect correlation (Figure 1D). Thus, our single-cell atlas provides a near-comprehensive transcriptional complement to the adult central brain connectome.

To uncover molecular features that distinguish neuronal cell types in the central brain, we performed Gene Ontology analysis of cell-type-defining genes (Figure S5D). We found that TFs and transmembrane receptors account for a large proportion of genes that distinguish neuronal cell types (Table S2). Many TFs show high cell-type specificity, consistent with their roles in establishing and maintaining neuronal identity. To identify those most critical for distinguishing neuronal cell types, we compared expression variability between and within cell types, identifying a subset that most strongly contributes to cell-type identity (Figure 1E). Unique combinations of these TFs can be used to define the complexity of cell types across the central brain (Figures 1F and 2; Tables S3 and S4). Many of these cell-type-defining TFs are established developmental regulators, supporting a model in which neuronal diversity arises from a combinatorial transcriptional code laid down during development.[12,30,31]

Among cell-type-defining TFs, domain enrichment analysis revealed a strong bias toward homeodomain-containing families (Figure 1G; Table S5), consistent with their well-established roles in patterning regional, segmental, and cell-specific identities in both vertebrate and invertebrate nervous systems.[32,33] The *Drosophila* central brain encompasses the cerebrum and gnathal ganglia. Each consists of three neuromeres, the morphological units along the body axis.[34] Previous work, including ours, has shown that the Hox homeodomain TFs, which define neuromeres, maintain their regional expression in adults.[35,36] We used expression from members of the Antennapedia complex of Hox genes to define cell types associated with the gnathal ganglia and tritocerebrum, while proto- and deutocerebrum-associated cell types were identified based on the absence of Hox gene expression, consistent with the anterior CNS being largely Hox negative[37,38] (Figures 1H–1J). In the ventral nerve cord (VNC), serially homologous cell types represent a conserved pattern of neuronal organization across segments.[39] The extent to which cell types are repeated or homologous within the brain and between the brain and the VNC is unclear. Anatomically comparing neuronal types and circuits to identify homologous structures is challenging due to the high degree of specialization in these two tissues. Our analysis suggests that leveraging single-cell transcriptomics will substantially advance future efforts to identify serially homologous cell types across the CNS.

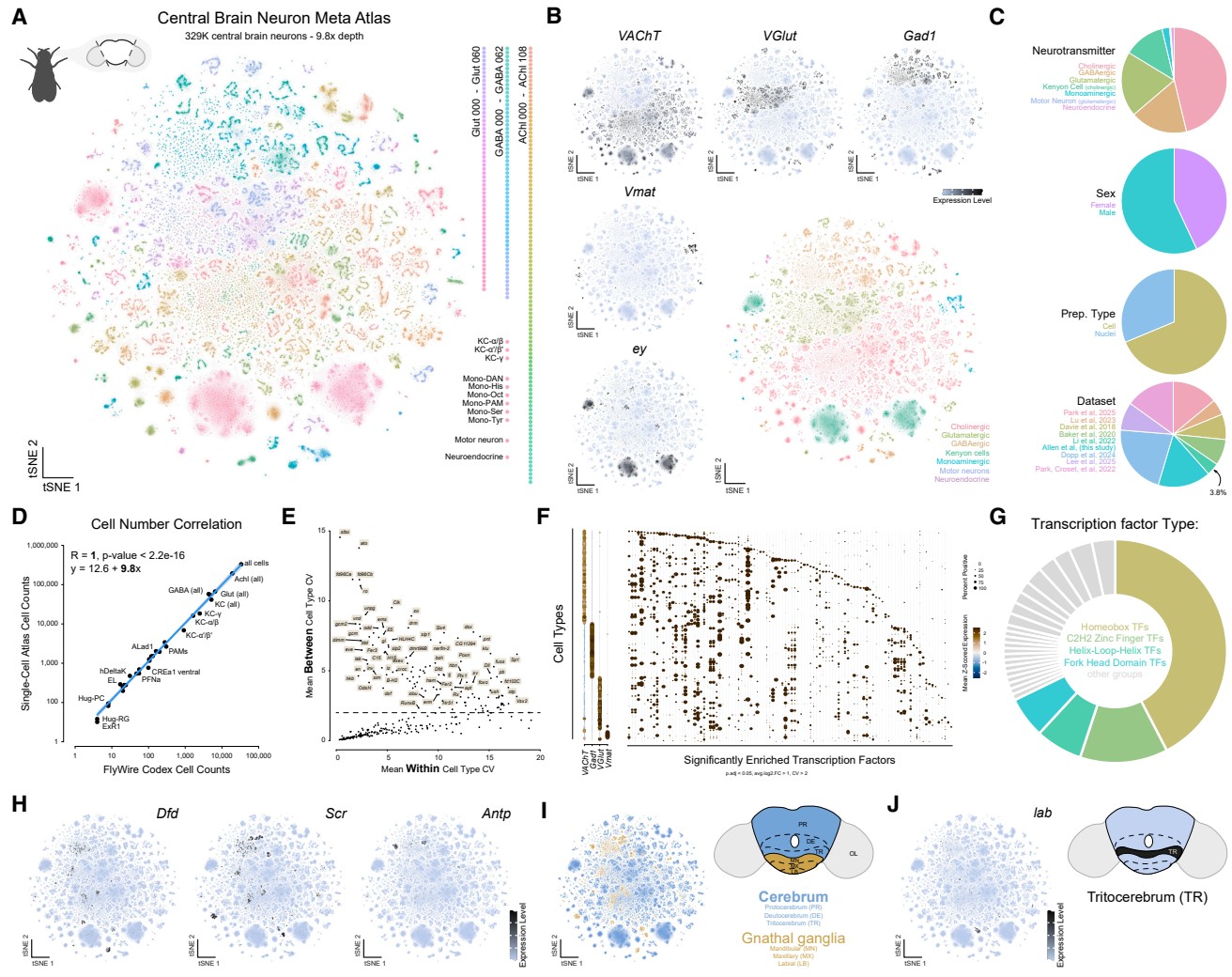

**Figure 1. A single-cell meta-atlas of *Drosophila melanogaster* central brain neurons**

(A) t-SNE of 329,466 central brain neurons, 9.8× cellular depth of coverage, colored by cell type.

(B) t-SNEs showing expression of fast-acting neurotransmitter biomarkers for acetylcholine (*VAChT*), glutamate (*VGlut*), and GABA (*Gad1*), as well as monoamines (*Vmat*) and Kenyon cells (*ey*).

(C) Pie charts representing the cellular composition of the central brain neuron atlas based on assigned broad cell type, sex, preparation type, and dataset origin.

(D) Correlation between single-cell atlas-derived cell counts and FlyWire-derived cell counts. Each point represents an anatomically defined neuronal cell type with a line of best fit ± 95% confidence intervals.

(E) Coefficient of variation (CV) analysis to identify transcription factors with high between-cell-type variability relative to within-cell-type variability. Each point represents a transcription factor.

(F) Dot plots of the expression of neurotransmitter identity (left) and significantly enriched transcription factors with the highest between-cell-type variability (right) across central brain neuronal cell types.

(G) Donut plot showing the distribution of significantly enriched transcription factors by protein domain family.

(H) t-SNEs of Hox gene expression (*Dfd*, *Scr*, and *Antp*) associated with the gnathal ganglia across the central brain.

(I) t-SNE (left) and schematic (right) annotating regions along the n-anterior/n-posterior neuraxis associated with the cerebrum (blue) and the gnathal ganglia (gold).

(J) t-SNE of *lab* expression (left) associated with the tritocerebrum with schematic (right).

See also Figures S1–S5 and Tables S1, S2, S3, S4, and S5.

## Transcriptionally defined cell types in the adult brain reflect developmental origins

To determine anatomical identities of transcriptionally defined cell types, we applied an intersectional genetic strategy (see STAR Methods) leveraging uniquely informative gene expression patterns, either individually or in combination (Figure 2A), to label neuronal populations (Figure 2B). These genetically defined types were then anatomically characterized and matched to their corresponding morphologies in the adult connectome (Figure 2C).[10,11] Using this approach, we repeatedly and consistently (8/8) found

**Figure 2. Hemilineages define transcriptional cell types in the adult central brain**

(A) Dot plot of cell-type-defining fast-acting neurotransmitter genes (*VAChT*, *Gad1*, and *VGlut*) and transcription factors (TFs; *drm*, *odd*, *TfAP-2*, *fd59A*, *dsf*, *bsh*, *trh*, *Awh*, *Ptx1*, and *Fer1*) across all neuronal clusters in the central brain.

(B) t-SNEs of highlighted cell types (black) associated with unique expression profiles (highlighted in A) within the neuronal central brain atlas, with zoomed-in insets (dashed box).

(C) Light microscopy (LM) images (top) showing neuronal populations identified via individual gene reporter (*odd*) or intersecting genes (*TfAP-2* and *Gad1*, *bsh* and *Gad1*, and *Ptx1* and *VGlut*). Electron microscopy (EM) reconstructions (bottom) of selected hemilineages from the FlyWire (DL2 ventral, CREa1 ventral, DM2 central, and SMPad1) based on transcriptionally defined cell types. LM images have been segmented (see Figure S6).

(D) UMAP of the SMPad1 hemilineage subclustered based on gene expression (left). Dot plot (right) of the expression of significantly enriched SMPad1 hemilineage-wide TFs (top) and subtype-specific TFs (bottom).

*(legend continued on next page)*

that intersected transcriptionally defined cell types represent hemilineages[29] (Figures 2C and S6), just as we and others had previously seen in the VNC.[35,40] For example, the co-expression in the protocerebrum of the TF *Ptx1* and the glutamatergic neuron marker *VGlut* corresponded to hemilineage SMPad1 (Figures 2C, bottom, and S6B–S6D). Thus, the overarching transcriptional relationship between neurons within the adult central brain reflects shared developmental origins.

As cell types within hemilineages are anatomically diverse, we assessed their transcriptional heterogeneity via a subclustering analysis of each of the 246 broad cell types of the central brain. We resolved 4,167 transcriptionally defined neuronal subtypes, providing an unprecedented view of adult brain transcriptional neuronal diversity (Figure S7). Similar magnitudes of neuronal diversity are seen in connectome-derived anatomical definitions of cell type.[10,11] To investigate the regulatory logic underlying this diversity, we focused on the SMPad1 hemilineage (Figures 2D–2K and S8). Subclustering analysis revealed 22 transcriptionally distinct subtypes, with hemilineage- and subtype-specific TF expression (Figure 2D). We used SCENIC[41,42] to infer gene regulatory networks (GRNs), integrating gene co-expression with DNA motif enrichment to predict TFs and their putative target genes (regulons; Figure 2E). Subclustering based on GRN-defined regulons closely matched clustering based on gene expression alone (Figure 2F), demonstrating that regulons robustly define hemilineage subtypes. We next examined regulons associated with hemilineage-defining TFs, including *Fer1*, *fkh*, *vvl*, and *Ptx1*. These regulons are highly interconnected, regulating large sets of functionally related downstream genes and forming extensive feedback loops between core regulators (Figures 2G–2I and S8B–S8D), suggesting coordinated control of subtype specification. Moreover, key neurodevelopmental genes within these regulons repeatedly emerged as markers of hemilineage subtypes when subclustering all transcriptionally defined neuronal cell types in the central brain (Figures 2J, 2K, and S7), indicating a shared regulatory architecture across diverse hemilineages.

## Neurodevelopmental gene expression distinguishes neuronal birth order across central brain subtypes

The most striking observation from our hemilineage analyses was the distinct spatial arrangements of cellular subtypes in our uniform manifold approximation and projection (UMAP), reflecting the transcriptional relationships among subtypes (Figures 2D and S7B). Some subtypes formed discrete, compact groupings resembling "punctate" clusters, representing cells with highly similar gene expression profiles. In contrast, other subtypes appeared to group together, forming elongated "serpentine" clusters, reflecting a transcriptional continuum between related subtypes. We examined their associated gene expression profiles to understand the molecular basis of these distinct cluster morphologies. We identified the neurodevelopmental gene *Imp* as highly enriched in subtypes exhibiting punctate morphology, while *dati* expression was highly correlated with serpentine subtypes (Figure 2K). *Imp* and *dati* are parts of a transcriptional temporal patterning cascade driving neurogenesis in the *Drosophila* CNS, with *Imp* associated with early-born neurons and *dati* associated with late-born neurons.[35,43–46] Thus, transcriptional relationships between punctate and serpentine subtypes appear to reflect neuronal birth order within hemilineages of the adult central brain.

To explore the broader relevance of birth order, we analyzed the relationship between *Imp* and *dati* expression across all neurons in the central brain. As we previously observed in the VNC,[35] their expression was largely mutually exclusive (Figures 3A–3C). *Imp*-enriched, early-born neurons accounted for approximately 30% of central brain neurons in our atlas (Figure 3D). Given that only 10% of the central brain in the connectome has been identified as embryonic-born primary neurons, this suggests that roughly two-thirds of the remaining *Imp*[+] early-born neurons arise post-embryonically and thus represent early-born secondary neurons (Figures 3D and 3E). Interestingly, *Imp*[+] early-born neurons consistently exhibited higher levels of total transcripts and total genes across all individual datasets included in our atlas (Figure 3F). These transcriptional differences suggest inherent distinctions in neuronal populations in the adult based on birth order.

To determine whether these differences reflect biological heterogeneity or technical artifacts, we employed three separate analytical strategies: (1) varying the number of principal components used in dimensionality reduction, (2) projecting early vs. late born into a shared embedding, and (3) independently subclustering each population. When we varied the number of principal components, *Imp* and *dati* remained mutually exclusive, but distinct cluster morphologies emerged only as we increased the number of principal components, and indeed, *Imp* and *dati* contribute to many principal components (Figure S9). Projecting early vs. late born into a shared embedding maintained both the mutual exclusivity and the punctate vs. serpentine t-distributed stochastic neighbor embedding (t-SNE) morphologies (Figures S10A and S10B). Finally, independently subclustering early-born and late-born neurons also preserved their respective t-SNE morphologies (Figures S10C and S10D). These findings suggest that differences in t-SNE morphologies are intrinsic

(E) UMAP of the SMPad1 hemilineage subclustered based on SCENIC GRN analysis (left). Dot plot (right) of the expression of significantly enriched SMPad1 hemilineage-wide regulons (top) and across hemilineage subtypes (below).

(F) Sankey diagram showing the agreement between mRNA-based and GRN-based clustering, with an adjusted Rand index (ARI).

(G) Network representation of SMPad1 hemilineage-identified GRNs; nodes are genes, edges are predicted regulatory interactions. Outer nodes are hemilineage-defining TFs (blue), and inner nodes are grouped and colored based on associated gene groups.

(H) Bar graph showing functional classification of regulon target genes based on gene groups.

(I) Subnetwork representation of SMPad1 main hemilineage-defining regulon TFs illustrating recurrent feedback and co-regulation.

(J) Subnetwork representation of SMPad1 hemilineage-defining regulons TFs (outer) and subtype target genes (inner), highlighting multiple known neurodevelopmental genes: *mamo*, *Eip93F*, *Imp*, *pros*, *br*, and *dati*.

(K) UMAPs of regulon target genes (*mamo*, *Eip93F*, *Imp*, *pros*, *br*, and *dati*) across SMPad1 hemilineage subtypes.

See also Figures S6–S8.

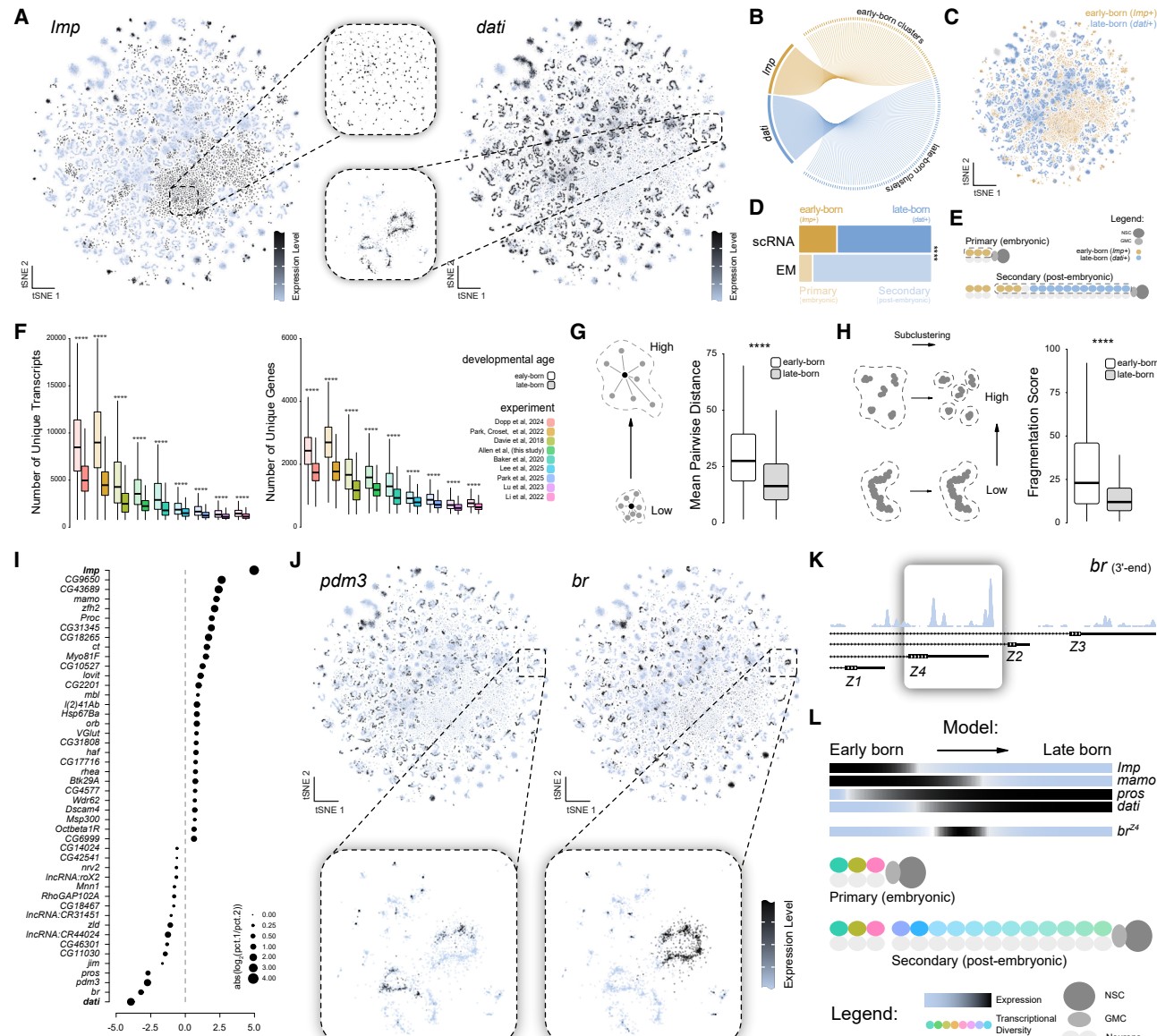

**Figure 3. Temporal patterning and transcriptional diversity in the central brain**

(A) t-SNEs showing the expression of *Imp* (left) and *dati* (right) across all central brain neurons (Kenyon cells removed). Insets (middle) highlight distinct clustering of *Imp*- and *dati*-expressing cells, reflecting their transcriptional and spatial organization.

(B) Chord diagram of *Imp* and *dati* and the clusters for which they are significantly enriched.

(C) t-SNE showing annotated early-born and late-born neurons across the central brain.

(D) Stacked bar plots comparing neuronal birth timing inferred from single-cell transcriptomics (scRNA) with lineage timing defined by electron microscopy (EM) (Bonferroni-corrected $\chi^2$ test, ****$p < 0.0001$).

(E) Schematic representing *Drosophila* hemilineage neurogenesis. Primary neurons (top dashed box) within a hemilineage are embryonic born, while secondary neurons (bottom dashed box) are post-embryonic born. *Imp* marks primary neurons and early-born secondary neurons, while *dati* marks only later-born secondary neurons.

(F) Boxplots showing the numbers of unique transcripts (left) and genes (right) in early-born and late-born neurons across individual datasets (Bonferroni-corrected Wilcoxon signed-rank test, ****$p < 0.0001$).

(G) Schematic (left) of mean pairwise distances among cells within a cluster (dashed line). Boxplot of mean pairwise distance across clusters split by $Imp^+$ neurons compared to $dati^+$ neurons (Bonferroni-corrected Wilcoxon signed-rank test, ****$p < 0.0001$).

(H) Schematic (left) showing that high fragmentation scores reflect more fragmented clusters (clusters within dashed lines). Boxplot (right) showing significantly higher fragmentation scores in $Imp^+$ neurons (Bonferroni-corrected Wilcoxon signed-rank test, ****$p < 0.0001$).

(I) Ranked dot plot of genes differentially expressed between $Imp^+$ and $dati^+$ neurons. Dot size represents the relative specificity of each gene's expression in $Imp^+$ vs. $dati^+$ neurons.

*(legend continued on next page)*

and robust features of the developmental stage-specific transcriptional complexity of early-born and late-born neurons rather than artifacts of dimensionality reduction.

To quantify transcriptional relationships within clusters, we calculated average pairwise distances, fragmentation scores, and modularity scores for cells expressing *Imp* vs. *dati* (Figures 3G, 3H, and S11). Early-born, *Imp*-enriched cells were more dispersed and fragmented, leading to their characteristic punctate t-SNE-morphology. In contrast, late-born, *dati*-enriched cells exhibited more connected transcriptional relationships, consistent with their serpentine t-SNE-morphology. Our findings reveal that *Imp* and *dati* not only mark neuronal birth order but also correlate with fundamentally distinct transcriptional profiles in the central brain.

To further explore differences between these neuronal populations, we identified genes most enriched in early- vs. late-born neurons. This analysis uncovered several neurodevelopmental genes indicative of neuronal birth order strongly associated with *dati* expression (Figure 3I). Notably, the TFs *Pdm3* and *br* were the most enriched in *dati*+ cells, with both genes displaying regional expression across multiple serpentine clusters in our atlas (Figure 3J). A specific isoform of *br*, Z4, known to mark cells born during larval L2–L3 ecdysis,[47] was the predominant isoform expressed in our atlas (Figure 3K). Cells expressing *br-Z4* consistently mark cells born at a specific developmental window across hemilineages, a pattern evident in our data. These observations support our broader model that transcriptionally defined cell types represent hemilineages, while subtypes reflect birth order. Moreover, within hemilineages, early-born subtypes are more transcriptionally distinct from one another and from late-born subtypes, suggesting that neurogenic timing not only shapes identity but also influences the granularity of transcriptional diversity within lineages (Figure 3L).

### Systematic reconstruction of birth order reveals repeated temporal transcription factor programs

Our findings suggest that transcriptional profiles in the adult brain retain signatures of developmental timing, enabling systematic reconstruction of neuronal birth order across hemilineages. As proof of principle, we investigated the anterodorsal antennal lobe (AL) olfactory projection neuron hemilineage, ALad1, whose birth order and transcriptional identities have been largely established.[48,49] We identified ALad1 neurons in our atlas, re-clustered, and transferred previously defined subtype annotations (Figures 4A–4C and S12; see STAR Methods). To investigate transcriptional temporal dynamics within the hemilineage, we performed pseudotime analysis, which orders cells along a continuous trajectory based on their gene expression profiles, anchoring the earliest-born neurons to *Imp* expression. This trajectory strongly correlated with experimentally validated birth order (Figure 4D), confirming that pseudotime recapitulates

temporal lineage progression in mature neurons and can be used to identify birth-order-associated transcriptional programs (Figure 4E).

Since unsupervised clustering in our atlas largely reflects hemilineage identity (Figure 2), we expanded pseudotime analysis to include multiple hemilineages (Figures 4F and 4G). We identified several tTFs repeatedly utilized across these selected hemilineages, reflecting conserved temporal developmental programs. To systematically examine birth-order-associated transcriptional dynamics, we extracted, re-clustered, and applied pseudotime analysis across most *dati*+ hemilineages (Figure 4H; see STAR Methods). By examining genes that vary across pseudotime within hemilineages, we can explore repetitive transcriptional signatures across the brain. Averaging the expression of the genes that varied over pseudotime confirmed the expected temporal expression patterns of *Imp*, *br*, and *dati*, representing early-, mid-, and late-born neurons, respectively (Figure 4I). Filtering for TFs with repeat occurrences across hemilineages revealed known and novel temporal regulators used throughout the central brain, highlighting shared mechanisms underlying neuronal birth order[9,14,47,50,51] (Figure 4J; Table S6). These findings indicate that temporal patterning mechanisms governing neurogenesis are deeply embedded in the adult transcriptional landscape across the central brain. As the two gene sets are mutually exclusive, these recurrent temporal regulators define an axis orthogonal to hemilineage-associated gene programs in shaping neuronal identity. Similar patterns were also observed in developing central brain lineages[52] and in the VNC.[53]

To explore the relationship between TF families and their roles in hierarchical neuronal identity, we compared the domain composition of TFs that define hemilineages (hTFs; Figure 1G) to those that vary with birth order (tTFs; Figure 4K). Hemilineage identity is predominantly specified by homeodomain-containing TFs, consistent with their conserved roles in establishing spatial and segmental identity across species.[54] In contrast, tTFs rely on a diverse set of domain classes, with a notable enrichment in Tramtrack-like (TTK-type) BTB-domain-containing TFs,[55] many of which are thought to act primarily as transcriptional repressors, potentially silencing alternative transcriptional programs to lock in subtype-specific neuronal fates.[52,56–58] These findings support a hierarchical model of transcriptional identity in which lineage-defining hTFs specify spatial origin, while temporal cascades of tTFs refine subtype fate over time. Interestingly, the TTK-type BTB domain family of proteins is arthropod specific and has undergone evolutionary expansion within modern insects.[59] Together, our findings align with evolutionary frameworks proposing that homeodomain TFs anchor conserved axial patterning programs, while expansion and diversification of multiple TF families underlie the emergence of neuronal subtype diversity and circuit elaboration across species.[33,52,60,61]

---

(J) t-SNEs (top) showing the expression of neurodevelopmental transcription factors *pdm3* (left) and *br* (right) across all central brain neurons. Insets (below) highlight the specific localization of these transcription factors within hemilineages.

(K) Isoform-specific expression (top) of *br* (Z1–Z4) within adult central brain neurons, showing high levels of Z4 isoform expression.

(L) Schematic model representing how temporal patterning during *Drosophila* neurogenesis (top) shapes transcriptional organization of neurons in the central brain. *Imp* marks early-born neurons with higher transcriptional diversity, while *dati* is enriched in late-born neurons. NSC, neuronal stem cell; GMC, ganglion mother cell.

See also Figures S9–S11.

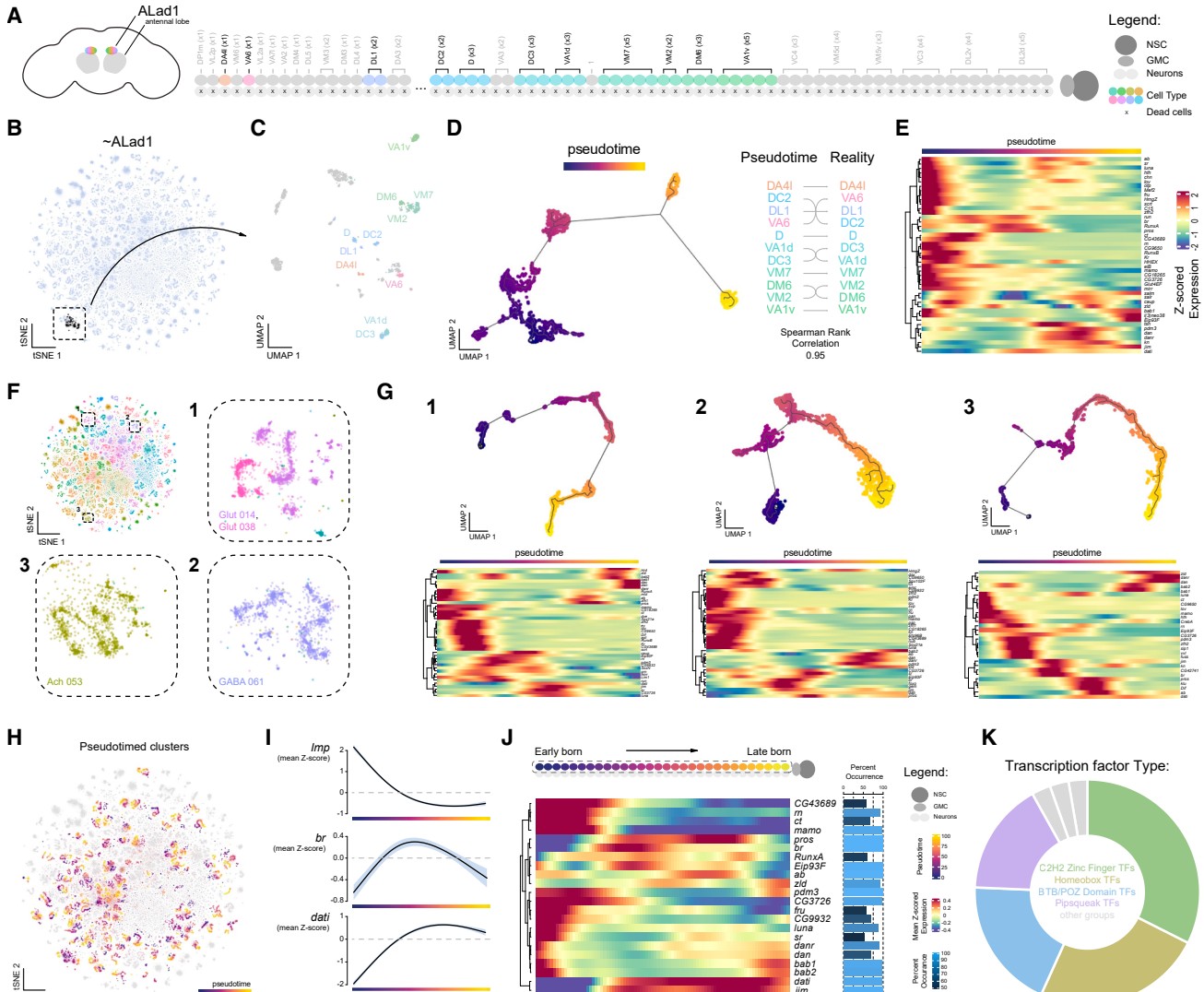

**Figure 4. A common set of repeated transcription factors defines transcriptional subtypes within hemilineages**

(A) Schematic of ALad1 hemilineage, arranged in developmental order (top); dead sister hemilineage below.

(B) t-SNE highlighting ALad1 neuronal cluster (black).

(C) UMAP of ALad1 hemilineage, with defined subtypes annotated (see STAR Methods).

(D) Pseudotime trajectory (left) compared to the real birth order of ALad1 cell types (right).

(E) Heatmap showing gene expression dynamics of transcription factors that significantly vary along pseudotime within the ALad1 hemilineage.

(F) t-SNE of the central brain atlas highlighting three distinct presumed hemilineage cell types for pseudotime analysis (1, 2, and 3) with zoomed-in regions.

(G) Pseudotime trajectories (top) for the three highlighted regions representing hemilineages. Heatmaps (bottom) showing gene expression changes of significantly enriched transcription factors across pseudotime.

(H) t-SNE of pseudotime-annotated neuronal clusters across the central brain, colored by pseudotime progression per cluster.

(I) Line plots showing mean Z-scored expression of key temporal markers (*Imp*, *br*, and *dati*) along the pseudotime axis, capturing their expected early, middle, and late transcriptional dynamics, respectively.

(J) Heatmap (left) averaging pseudotime-ordered gene expression across central brain clusters identified novel transcription factors of neurodevelopmental windows that repeat across cell types within hemilineages. Bar plot (right) of the percentages of lineages where these factors varied significantly across pseudotime.

(K) Donut plot showing the distribution of transcription factors that vary based on birth order across the central brain by protein domain family.

See also Figure S12 and Table S6.

## Uncovering physiological properties through cell-type annotation of the central brain

To further resolve molecular and physiological identities of neurons in the adult central brain, we annotated neuronal cell types

utilizing existing single-cell and bulk RNA-seq datasets. Neurons in the central brain are primarily derived from one of two types of neuroblasts—type I and type II (Figure 5A; reviewed in Holguera and Desplan,[12] Sousa-Nunes et al.,[62] Homem et al.,[63]

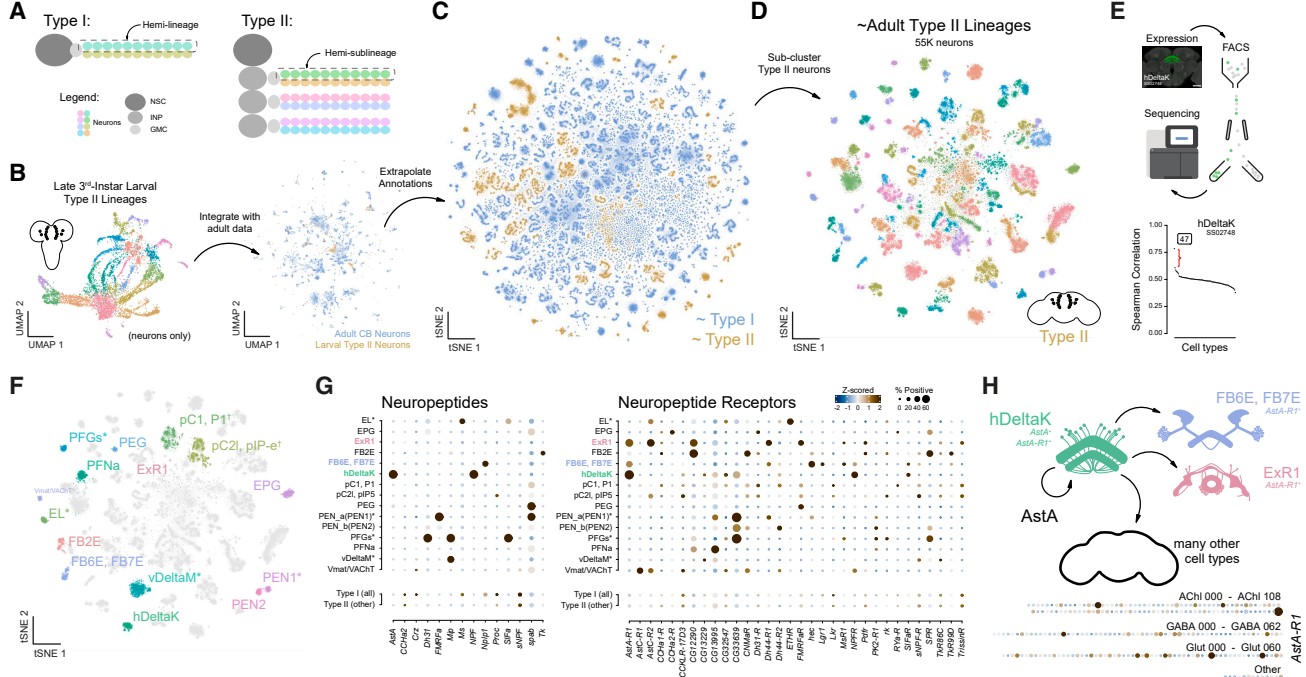

**Figure 5. Integrative analysis of type II lineages in the central brain**

(A) Schematic illustrating type I hemilineages and type II hemi-sublineages.

(B) UMAPs of larval type II data (left) integrated with adult central brain data (right). Reprocessed larval type II data were originally from Michki et al.[65] and Rajan et al.[66]

(C) t-SNE of the central brain neuronal atlas with extrapolated annotations highlighting neuroblast lineage types—type I (blue) and type II (gold).

(D) t-SNE of ~55,000 type II neurons subclustered.

(E) Workflow for annotating cell types from FACS bulk sequencing data: targeted expression based on genetic access, FACS, and RNA sequencing (top). Spearman correlation analysis of cell types in the adult type II atlas compared to hDeltaK sequencing data (bottom). Immunostaining of hDeltaK (SS02748) was obtained from FlyLight. Bulk RNA-seq data were originally from Wolff et al.[68]

(F) Annotated t-SNE of type II atlas (see Figures S10 and S11 and STAR Methods).

(G) Dot plot of neuropeptide (left) and neuropeptide receptor (right) expression across annotated type II cell types.

(H) Schematic of AstA signaling in hDeltaK neurons and AstA-R1-expressing cell types. hDeltaK neurons (*AstA+* and *AstA-R1+*) and other *AstA-R1+* central brain cell types (e.g., FB6E, FB7E, and ExR1) likely represent localized AstA-modulated circuits. A dot plot of *AstA-R1* expression across the central brain reveals a broader network of potential signaling partners.

See also Figures S13 and S14.

and Doe.[64]). To distinguish lineage origin, we reprocessed and integrated published larval type II scRNA-seq data[65,66] with our adult atlas (Figure 5B), enabling systematic assignment of adult neurons to type I or type II neuroblast lineages (Figure 5C). This analysis reinforced the lineage-based organization of transcriptionally defined cell types and confirmed that previously identified hemilineages correspond to their expected neuroblast types (Figure 2C). We re-clustered all type II neurons, generating an adult type II neuroblast lineage atlas (Figures 5D and S13A). A recently published fluorescence-activated cell sorting (FACS)-based study independently produced a comparable atlas,[67] providing complementary validation and reinforcing a shared framework for understanding type II lineage organization.

To map transcriptionally defined clusters in our type II atlas to anatomically defined cell types, we leveraged previously generated bulk RNA-seq datasets of genetically labeled and fluorescence-activated cell-sorted central brain populations[68–70] (Figures 5E and S13B–S13E). Correlating these profiles with our atlas (see STAR Methods) enabled high-confidence annotation of type II cell types based on distinct transcriptional signatures (Figures 5F and S13B–S13E). Additionally, we leveraged genetic intersection[29] and neuropeptide (NP) expression as molecular markers[68] to identify central brain cell types, resulting in a molecular map of many type-II-derived cell types in the adult brain (Figures 5F, 5G, and S14).

To highlight the functional relevance of our annotations, we examined NP and NP-receptor expression across the type II atlas (Figures 5G, 5H, S14D, and S14E). In agreement with prior studies,[68] *AstA* expression was enriched in hDeltaK neurons, while its receptor, *AstA-R1*, was detected in FB6E, FB7E, ExR1, and hDeltaK neurons, suggesting local and autoregulatory *AstA* signaling within the central complex, a hub for navigation and behavioral state control[71,72] (Figures 5G and 5H). Interestingly, hDeltaK neurons had the highest expression of *AstA* and *AstA-R1* among all cell types across the central brain, suggesting that autocrine signaling within this population facilitates a homeostatic feedback loop (Figures 5H and S14E). Additionally, we identified multiple unannotated *AstA-R1+* cell types

distributed across the central brain, highlighting new targets of *AstA* signaling and expanded roles for hDeltaK neurons in modulating both local and long-range circuits (Figure 5H, bottom). Our atlas provides a framework for targeted genetic access to these neuronal populations and enables mechanistic dissection of neuropeptidergic regulation of central-complex-dependent behaviors such as navigation or arousal-related activity states.

### Transcriptional and functional specialization of neuroendocrine cells in the central brain

Overall, cells tend to cluster by hemilineage in unsupervised analyses of gene expression, reflecting their shared developmental origins. However, this was not always the case, as some cells cluster due to convergent gene expression patterns, reflecting shared physiological properties. This pattern is especially evident in neuroendocrine neurons, which secrete NPs to regulate homeostatic and behavioral processes in response to internal state and environmental cues.[73] As shown previously,[35] the TF *dimm*, essential for the differentiation of neurosecretory cells,[74–76] enables robust identification of neuroendocrine populations in the adult brain (Figure 6A). Isolating *dimm*+ cells and subclustering based on gene expression and GRNs revealed 34 highly transcriptionally distinct neuroendocrine subtypes (Figures 6B, 6C, and S15A). Consistent with their embryonic origin, all neuroendocrine subtypes expressed *Imp* (Figure 3A). Similarly, despite their distinct developmental origins,[10,11] monoaminergic cells clustered together within our atlas (Figures S16 and S17).

To probe the functional diversity of neuroendocrine subtypes, we examined NP expression, associating specific NPs with each subtype (Figure 6D). Over half of the neuroendocrine subtypes showed weak or no detection of fast-acting neurotransmitter or monoaminergic neuron markers, suggesting they are specialized for NP release. Despite their shared functional role in NP release, neuroendocrine cell types arise from different hemilineages, giving each subtype distinct transcriptional signatures. Moreover, SCENIC-based regulon analysis further confirmed broad regulatory heterogeneity, revealing diverse GRN usage across subtypes (Figure 6E). While neuroendocrine neurons constitute a small fraction of the central brain, many additional neurons express NPs at lower levels, likely supporting local signaling.[77] We classified NPs into four categories based on these expression patterns—very broad, broad, restricted, and very restricted—revealing that some NPs show broad expression within, but restricted to, specific neurotransmitter cell types (e.g., *spap* with acetylcholine and *Nplp1* with glutamate/GABA) (Figure S15C). In contrast, others (e.g., *Orcokinin* and *Ilp2*) are restricted to a few neuroendocrine neurons. These findings highlight NP signaling's transcriptional and functional diversity in the central brain, reflecting their specialized roles in coordinating physiological and neuronal processes.

To demonstrate that our atlas reliably links transcriptional identity with anatomical identity—even for rare cell types—we examined neuroendocrine subtypes expressing the NP Hugin (Hug). Hug neurons have been shown to integrate internal state and sensory cues, modulating behaviors such as feeding, locomotion, and circadian activity.[78–82] We focused on the homeodomain TF Otp and its regulon, as SCENIC analysis predicts

that Otp regulates Hug, and its regulon showed restricted expression across neuroendocrine subtypes, particularly in three Hug-expressing subtypes (Figures 6F and 6G). Most genes identified within the *otp* regulon were significantly enriched as markers for all Hug-expressing subtypes (Figure 6F). Using a combination of cell number representation and HOX gene expression (e.g., *Dfd*), we linked transcriptional subtypes to anatomically defined subtypes in FlyWire (Figures 6H–6J). By examining specifically enriched genes for different Hug-expressing subtypes (Figure 6K), we uncovered new insights into their functional properties, including the co-release of other NPs and their potential to respond to distinct monoamines (e.g., *Octbeta1R* in Hug-RG neurons). Importantly, these neurons constitute rare cell types—four cells per brain—demonstrating that our atlas possesses sufficient resolution to identify unique transcriptional profiles and functional characteristics of rare neuronal populations. As the necessity for cell-specific genetic access becomes increasingly important, our atlas serves as a valuable resource for disentangling neuronal heterogeneity present in existing tools, even for rare cell types, enabling more precise and interpretable behavioral studies in the future.[83]

### An interactive resource for exploring the *Drosophila* central brain atlas

Our analysis, with its unprecedented resolution, provides an invaluable framework for interpreting single-cell data from the *Drosophila melanogaster* central brain, enabling the exploration of relationships between developmental origins, timing, and functional identities (Figure 7A). We define a logic for genetically accessing specific neuronal cell types by combining neurotransmitter identity markers with hemilineage-restricted (hTFs) and temporally restricted TFs (tTFs), providing a tractable framework for targeting transcriptionally distinct neuronal subtypes across the central brain (Figure 7B). To enhance accessibility and usability, we have deployed a user-friendly website (https://www.flycns.com), offering interactive web-based visualization of the atlases generated in this study (Figure 7C). The site includes broad atlases, such as those covering the whole head, whole head neurons, and central brain neurons. Additionally, specific subatlases are available for distinct neuronal populations, including adult type II neurons, early-born and late-born neurons, neuroendocrine cells, and monoaminergic neurons. More specialized subclustering atlases, such as those focused on individual hemilineages, are also available. Each atlas includes both known and predicted cell type annotations. Importantly, this intuitive interface enables easy visualization of single-cell transcriptomic data for the broad research community to explore cellular diversity and transcriptional complexity within the *Drosophila* central brain (Figure 7C). This study acts as an essential companion to the electron microscopy (EM)-resolution maps of the *Drosophila* central brain,[10,11] as integrating neuronal connectivity with molecular identity will enable a mechanistic understanding of how circuit structure links to function and ultimately behavior.

### DISCUSSION

Single-cell transcriptomic profiling technologies have revolutionized reductionist approaches to studying complex biological

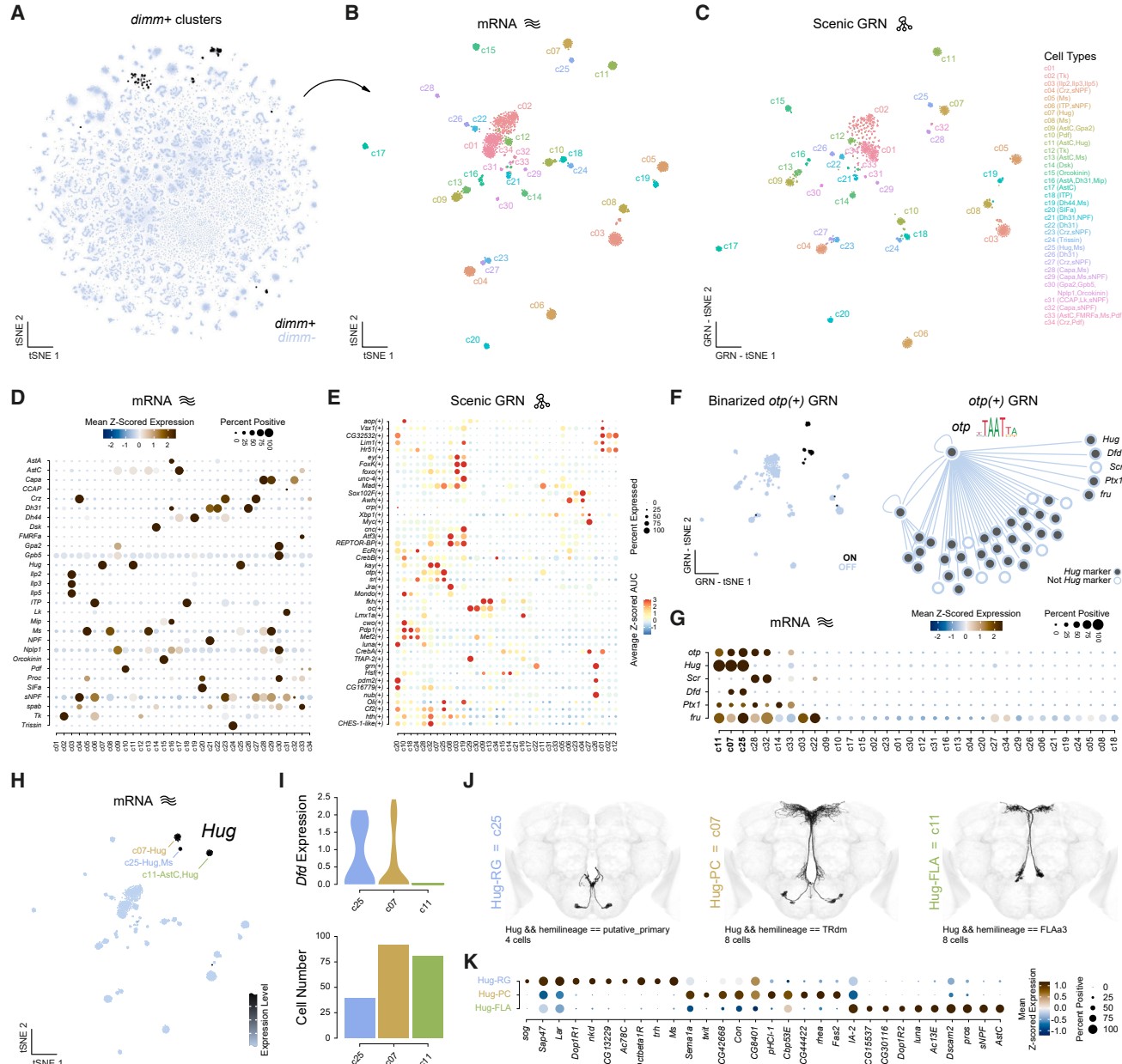

**Figure 6. Transcriptional diversity of central brain neuroendocrine cell types**

(A) t-SNE highlighting neuropeptide-expressing cell types based on the expression of *dimm*.

(B) t-SNE of subclustered neuroendocrine cell types.

(C) GRN-t-SNE of subclustered neuroendocrine cells based on SCENIC GRN analysis.

(D) Dot plot of neuropeptide expression across neuroendocrine subtypes.

(E) Dot plot of significantly enriched transcription factor regulons identified by SCENIC GRN analysis across neuroendocrine subtypes.

(F) Binarized regulon activity of the transcription factor *otp* in neuroendocrine subtypes (left). GRN showing *otp* driving the expression of multiple genetic markers for *Hug*⁺ cell types (right).

(G) Dot plot of *otp*-regulated genes (*Hug*, *Scr*, *Dfd*, *Ptx1*, and *fru*) across neuroendocrine subtypes.

(H) t-SNE showing *Hug* expression in neuroendocrine subtypes, highlighting *Hug*-expressing subtypes with other co-expressed NPs.

(I) Violin plots of *Dfd* expression levels across *Hug*-expressing subclusters (top). Bar plot showing the number of cells in each subcluster (bottom).

(J) EM reconstructions of the three *Hug*-expressing neuroendocrine subtypes in the central brain from the FlyWire dataset. Anatomical and transcriptional subtypes have been correlated based on *Dfd* expression and cell number representation.

(K) Dot plot of the top genes defining *Hug*⁺ subtypes.

See also Figures S15–S17.

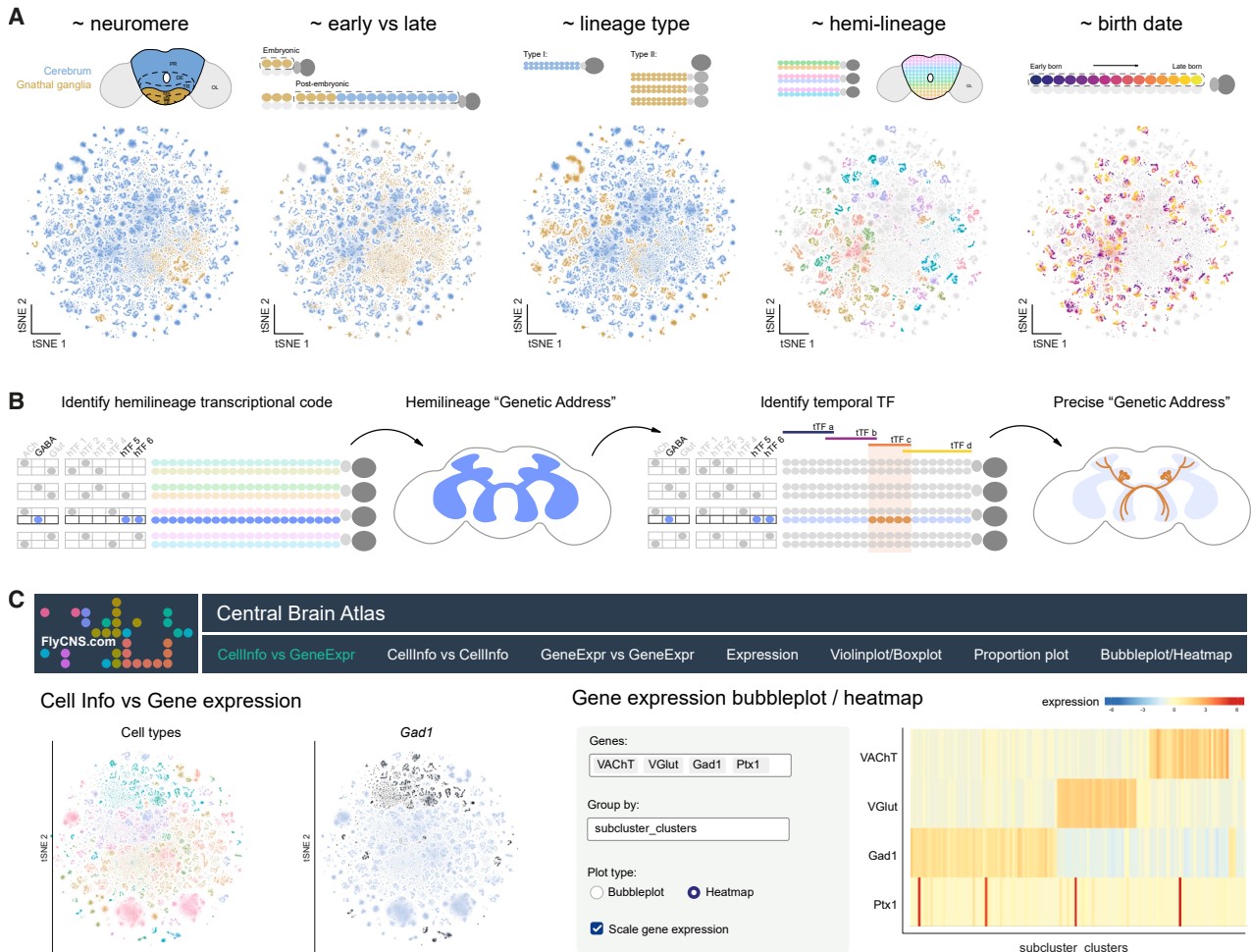

**Figure 7. Interactive exploration of the central brain neuronal atlas in *Drosophila***
(A) Organizational principles of the central brain neuron atlas. Schematics and t-SNEs of the central brain neuron meta-atlas (Kenyon cells removed) classified across multiple analyses.
(B) Schematic for defining precise "genetic addresses" for neuronal cell types in the adult central brain by intersecting hemilineage-restricted transcription factors (hTFs) with temporally restricted transcription factors (tTFs) associated with neuronal birth order.
(C) Overview of the interactive web-based platform for exploring the central brain neuron atlas, available at flycns.com.

systems. Constructing comprehensive transcriptional atlases for complex organisms like *Drosophila melanogaster* is now an attainable goal, and the *Drosophila* research community has made substantial progress in this endeavor – most notably through the FCA.[25] Nevertheless, the diversity and complexity of cell types vary significantly across tissues, posing unique challenges. Among these, the central brain—the fly's higher-order processing and integration center that underlies its rich behavioral repertoire—is one of the most complex, owing to its vast array of highly specialized neuronal cell types. As a result, it remains underrepresented in current resources. We address this gap by generating a high-resolution single-cell transcriptional atlas of the *Drosophila melanogaster* central brain, identifying 4,167 distinct neuronal subtypes derived from 246 broader types (Figure S7). Integrating newly generated and existing single-cell datasets offers a scalable and cost-effective strategy

for optimizing available resources. This work represents a significant step forward in capturing the cellular diversity of the *Drosophila melanogaster* central brain, providing an invaluable resource for the research community.

Our unbiased investigation into the transcriptional characteristics defining neuronal cell types in the adult central brain revealed that shared developmental histories overwhelmingly emerge as the dominant organizing principle. Transcriptionally defined cell types correspond closely to neuroblast-derived hemilineages (Figure 2). During development, these lineages are specified by unique combinations of TF expression—some of which persist into adulthood—providing each lineage with a distinct "genetic address" (Figure 7B).[84] This observation suggests that a neuron's developmental origin exerts a stronger influence on its mature molecular identity than previously appreciated. Our findings in the brain parallel those from our earlier study on the adult VNC,[35]

where many of our predictions regarding hemilineage transcriptional identities have since been validated.[40] This convergence across both major motor and integrative centers supports the idea that lineage-based transcriptional identities remain a fundamental axis of neuronal organization in *Drosophila*—and likely in other species with stereotyped neurogenesis. Retaining lineage-based identities likely constrains diversification via temporal patterning, sexual differentiation, and plasticity, shaping how circuits diversify within individuals and evolve across species while preserving core architecture.

A further defining feature, apparent only in our high-resolution atlas, is the striking transcriptional distinctness between early-born and late-born neurons. Neuroblasts progress through a series of TF expression windows that govern the diversity of neurons they generate. This transcriptional progression acts as a temporal recorder of neuroblast divisions, with *Imp* marking neurons from early stages and *dati* marking neurons from later stages. In our atlas, early-born (*Imp*$^+$) neurons and late-born (*dati*$^+$) neurons form transcriptionally distinct classes (Figures 3 and S9–S11), with early-born neurons exhibiting more complex and distinct transcriptional profiles compared to late-born neurons, which often share overlapping profiles within their hemilineages. Early-born neurons have been found to have uniquely elaborate projection patterns and are proposed to serve as pioneering neurons, critical for establishing larval or adult brain architecture.[85,86] In contrast, late-born neurons integrate into preexisting circuits, increasing the diversity of cell types. Thus, differing transcriptional complexities may reflect these differing developmental demands. Furthermore, in the recent adult brain connectome, it was found that known embryonically born neurons have significantly larger cell bodies and cell body fibers and often were unable to reliably be assigned to a lineage given their often-unique morphologies, indicating that their unique identities remain in the adult.[10,11] We show that genetic intersections that capture entire hemilineages can be deduced from our single-cell atlas, thereby integrating anatomical structures with their lineage-based developmental origins in the central brain.

As developmental timing appeared to shape transcriptional relationships among neurons in the adult brain, we systematically reconstructed neuronal birth order across pseudotime within hemilineages across the central brain. We identified tTFs expressed during development that persist into adulthood, reinforcing a developmentally rooted recurrent tTF code that underlies much of the cellular diversity in the central brain (Figure 4). For example, the Mamo > Pdm3 > Br-Z4 > Ab > Bab1/2 > Pdm3 temporal cascade identified in our analysis highlights how developmental mechanisms governing temporal transitions during neurogenesis ultimately shape hemilineage subtypes in the adult brain, as previously described in the embryo.[14] Therefore, TF expression not only identifies transcriptionally unique cell types (Figure 2) but also allows for refined subclustering of hemilineages (Figures 4 and 7B; see Allen et al.[29]), expanding the resolution to thousands of distinct neuronal types (Figure S7) and reflecting the massive diversity of the central brain. Overall, our findings support the neuronal lineage temporal patterning model previously proposed,[85] where neuronal complexity decreases over time across hemilineages through a progressive code of broad TF expression across early- to late-born neurons, while recurring TFs and a mix of additional TFs combine to diversify temporal windows. This principle is conserved across phyla, including in vertebrate systems such as the spinal cord and retina, where sequential gene expression in progenitors similarly governs neuronal subtype identity.[30,87–90] These data show that temporal patterning mechanisms governing neurogenesis are deeply embedded in the adult transcriptional landscape. Moreover, the use of distinct TFs along spatial (hemilineage) and temporal (birth order) axes suggests a modular strategy for generating neuronal diversity, consistent with principles proposed in bilaterian nervous system evolution.[33,60,61]

While developmental histories broadly define transcriptional identities, our atlas also identified neuronal cell types that exhibit transcriptional convergence due to shared physiological roles despite diverse developmental origins. This difference is particularly evident in neuroendocrine and monoaminergic neurons, both of which are sparsely located throughout the brain, yet are disproportionately influential in orchestrating brain-wide states. In these cases, a shared function, neuromodulation via NP or monoamine release, respectively, overrides lineage-derived transcriptional signatures (Figures 6 and S15–S17). The balance between developmental constraints and functional convergence may represent an important route by which central brain circuits acquire new capabilities while retaining their lineage-based organization, offering insights into how complex behaviors evolve.

Given that much of the central brain remains unannotated, our atlas and analytical approaches lay the foundation for systematically identifying and classifying all transcriptionally distinct cell types within the central brain and linking them to their anatomical counterparts. Our dataset categorizes neurons based on multiple genetically identifiable characteristics, including neuromere location, neuroblast origin, birth order, and physiological properties such as neurotransmitter expression. The transcriptional identity of central brain lineages has remained poorly defined. This study provides a critical schema for the systematic annotation of hemilineages, allowing for a more complete understanding of the brain's developmental architecture. Additionally, we present a strategy for linking transcriptionally defined cell types to their anatomical identities via genetic intersections, which can then be mapped onto the connectome (Figure 2). This can even uncover extremely rare neuron types whose behavioral roles can now be probed through targeted, testable hypotheses. Furthermore, we demonstrate that transcriptionally and anatomically defined cell types can now be reinterpreted in the context of the entire central brain (Figure 5), enabling detailed comparisons between neuronal cell types. Integrating transcriptional profiles with the connectome provides a powerful approach for predicting how neurons communicate based on the genes they express. Indeed, our analyses suggest that transcriptomic identity and anatomical (connectome-based) classification represent equally informative and distinct axes for defining cell types in the brain. These insights will feed directly into systems neuroscience models, offering valuable information about the brain's operation.

This study defines core heuristics for classifying neuronal identity in the central brain, offering a blueprint for integrating

transcriptional, anatomical, and functional modalities (Figure 7). Current single-cell technologies face limitations in detecting more subtle, state-dependent changes in neuronal gene expression. However, the increasing resolution of these methods and the ability to integrate new datasets with our atlas hold promise for addressing these challenges in the future. Our findings underscore the importance of linking adult neuronal characteristics to their developmental origins. Future work exploring the developmental dynamics that drive central brain neuronal diversity will be crucial for understanding how transcriptional identity influences adult neuronal connectivity, physiology, and behavior. Finally, our atlas also sets the stage for evolutionary comparisons, enabling cross-species analysis of neuronal lineages, molecular programs, and circuit architectures. Identifying conserved transcriptional programs, such as tTF cascades, will be key to understanding how nervous systems evolve to meet species-specific ecological demands.

### Limitations of the study

While integrating multiple, diverse single-cell datasets demonstrates that neuronal identities are robust across genetic backgrounds and physiological states, this approach may mask more subtle transcriptional differences between cell types. Future studies with deeper coverage and standardized genetic backgrounds across biological replicates will be important for resolving transcriptionally similar subtypes. Furthermore, our reliance on 10× Genomics' 3′ scRNA-seq technology limits transcript coverage and sensitivity. Because only the 3′ ends of transcripts are captured, full-length isoforms, alternative splicing events, and low-abundance transcripts may be underrepresented. As a result, some transcriptional complexity may be underestimated. Future studies using long-read sequencing will provide a more comprehensive view of isoform diversity and splicing. Finally, while our intersectional strategies offer strong genetic access to specific hemilineages and subtypes, full functional validation of these tools is ongoing. In addition, although our analyses distinguish between hemilineage-defining and temporally patterned TFs, these classifications are based on adult transcriptomic data. Definitive confirmation will require longitudinal developmental time series and *in vivo* functional studies to determine causal roles in cell-type specification and maintenance.

### RESOURCE AVAILABILITY

#### Lead contact
Requests for further information, resources, and reagents should be directed to and will be fulfilled by the lead contact, Stephen F. Goodwin (stephen.goodwin@cncb.ox.ac.uk).

#### Materials availability
All unique/stable reagents generated in this study are available from the lead contact without restriction.

#### Data and code availability
Datasets described here can be visualized at https://www.flycns.com/. Sequencing files, digital expression matrices, and Seurat RDS files are available from the Gene Expression Omnibus (GEO; https://www.ncbi.nlm.nih.gov/geo/), accession number GSE296540. The code used in this analysis is available from GitHub (https://github.com/aaron-allen/Dmel-adult-central-

brain-atlas) and Zenodo (https://doi.org/10.5281/zenodo.17513514). Light microscopy images are available at Virtual Fly Brain (https://v2.virtualflybrain.org). Previously published data used in this study are listed in the key resources table.

### ACKNOWLEDGMENTS

We thank E.J. Clowney, Y. Ding, E. Rideout, S. Russell, D. Shepherd, S. Waddell, J. Walsh, and members of the Goodwin lab for inspiration, insightful discussions, and critical reading of the manuscript. We also thank F. Casares, G. Miesenböck, and T. Shirangi for generously providing stocks and reagents. We are grateful to S. Liu for providing early access to data from Dopp et al.[20] Stocks obtained from the Bloomington Drosophila Stock Center (NIH P40OD018537) were used in this study. We thank A. Rings, S. Birtles, V. Croset, and C. Treiber for assistance with dissection and dissociation. Sequencing was performed by the Oxford Genomics Centre. A.M.A. and M.C.N. were supported by a Wellcome Trust Senior Investigator Award to S.F.G. (106189/Z/14/Z) and by a BBSRC grant (BB/X016595/1) to M.C.N., A.M.A., and S.F.G. T.N. and F.A. were supported by a BBSRC grant (BB/Y001869/1) to T.N. and S.F.G. A.M.A. and D.A. were supported by a Wellcome Trust Collaborative Award to D.S. and S.F.G. (209235/Z/17/Z).

### AUTHOR CONTRIBUTIONS

Conceptualization, A.M.A., M.C.N., and S.F.G.; methodology, A.M.A., M.C.N., T.N., F.A., D.A., D.S., and S.F.G.; investigation, A.M.A., M.C.N., T.N., D.A., and F.A.; resources, S.F.G.; writing, A.M.A., M.C.N., T.N., F.A., and S.F.G.; supervision, D.S. and S.F.G.; funding acquisition, D.S. and S.F.G.

### DECLARATION OF INTERESTS

The authors declare no competing interests.

### DECLARATION OF GENERATIVE AI AND AI-ASSISTED TECHNOLOGIES IN THE WRITING PROCESS

During the preparation of this work, the authors used ChatGPT, Google Gemini, and Grammarly to assist with code and grammatical edits of text. The authors reviewed and edited the content as needed and take full responsibility for the publication's content.

### STAR★METHODS

Detailed methods are provided in the online version of this paper and include the following:

- KEY RESOURCES TABLE
- EXPERIMENTAL MODEL AND STUDY PARTICIPANT DETAILS
  - Drosophila stocks
  - Full genotype list
- METHOD DETAILS
  - Central brain single-cell sample preparation
  - Single-cell RNA-seq using 10x chromium
  - Alignment and cell identification
  - Post-alignment processing
  - Re-processing of publicly available scRNA-seq data
  - Integrating datasets
  - Evaluating other integration methods
  - Annotation of sex
  - Annotation of broad cell types
  - Combinatorial cell type markers
  - Gene ontology analysis
  - Gene regulatory network analysis with SCENIC
  - Early- vs. late-born morphology quantification
  - Annotating and subclustering ALad1
  - Pseudotime analysis
  - Annotating and subclustering type II NSC lineages

  ○ Annotating and subclustering neuroendocrine and monoaminergic neurons
  ○ Generating a web app
  ○ Generation of split-Gal4 driver lines
  ○ Immunohistochemistry
  ○ Confocal image acquisition and processing

### SUPPLEMENTAL INFORMATION

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

## STAR★METHODS

### KEY RESOURCES TABLE

| REAGENT or RESOURCE | SOURCE | IDENTIFIER |
|---|---|---|
| **Antibodies** | | |
| goat anti-chicken Alexa Fluor 488 | Thermo Fisher Scientific | Cat# A-11039*; RRID:*AB_2534096 |
| goat anti-rabbit Alexa Fluor 488 | Thermo Fisher Scientific | Cat# A-11034; RRID:AB_2576217 |
| goat anti-mouse Alexa Fluor 546 | Thermo Fisher Scientific | Cat# A-11030*; RRID:*AB_2534089 |
| goat anti-rabbit Alexa Fluor 546 | Thermo Fisher Scientific | Cat# A-11035*; RRID:*AB_2534093 |
| goat anti-mouse Alexa Fluor 633 | Thermo Fisher Scientific | Cat# A-21050**; RRID:*AB_2535718 |
| anti-GFP polyclonal (rabbit) | Thermo Fisher Scientific | Cat# A-6455*; RRID:*AB_221570 |
| anti-Brp (nc82) monoclonal (mouse) | Developmental Studies Hybridoma Bank | Cat# nc82; RRID:AB_2314866 |
| anti-RFP (rabbit) | Antibodies Online | Cat# ABIN129578; RRID:AB_10781500 |
| anti-GFP (chicken) | Abcam, UK | Cat# ab92456*; RRID:*AB_10561923 |
| **Chemicals, peptides, and recombinant proteins** | | |
| Calcium- and Magnesium-free DPBS | Gibco™ | Cat# 14190086 |
| Schneider's *Drosophila* medium | Gibco™ | Cat# 21720–001 |
| Tetrodotoxin | Abcam, UK | Cat# ab120054 |
| D(−)-2-Amino-5-phosphonovaleric acid | Sigma-Aldrich | Cat# A8054 |
| 6,7-dinitroquinoxaline-2,3-dione | Sigma-Aldrich | Cat# D0540 |
| Papain | Sigma-Aldrich | Cat# P4762 |
| Collagenase I | Sigma-Aldrich | Cat# C2674 |
| Formaldehyde | Sigma-Aldrich | Cat# 47608-250ML-F |
| Phosphate-buffered saline (PBS) | Sigma-Aldrich | Cat# P3183-10PAK |
| Triton X-100 | Sigma-Aldrich | Cat# T8787-100ML |
| Normal Goat Serum | Sigma-Aldrich | Cat# G9023 |
| Vectashield mounting medium | Vector Laboratories | Cat# H-1000, RRID; AB_2336789 |
| **Critical commercial assays** | | |
| Chromium™ Single Cell 3' Library Kit v2 | 10x Genomics | Cat # 120234 |
| **Deposited data** | | |
| Central brain scRNA-seq | This study | GEO: GSE296540 |
| Whole brain scRNA-seq | Davie et al., 2018[17] | GEO: GSE107451 |
| Whole brain scRNA-seq | Baker et al., 2021[18] | GEO: GSE152495 |
| Central brain scRNA-seq | Park et al., 2022[19] | GEO: GSE207799 |
| Central brain scRNA-seq | Dopp et al., 2024[20] | GEO: GSE221239 |
| Central brain snRNA-seq | Lee et al., 2025[21] | GEO: GSE247965 |
| Optic lobe scRNA-seq | Ozel et al., 2021[22] | GEO: GSE142789 |
| Optic lobe scRNA-seq | Kurmangaliyev et al., 2020[23] | GEO: GSE156455 |
| Adult retina scRNA-seq | Yeung et al., 2022[24] | GEO: GSE214510 |
| Fly Cell Atlas, head snRNA-seq | Li et al., 2022[25] | ENA: E-MTAB-10519 |
| Aging Fly Cell Atlas, head snRNA-seq | Lu et al., 2023[26] | GEO: GSE218661 |
| Alzheimer's disease model Fly Cell Atlas, head snRNA-seq | Park et al., 2025[27] | GEO: GSE261656 |
| Larval type II lineages | Michki et al., 2021[65] | GEO: GSE153723 |
| Larval type II lineages | Rajan et al., 2023[66] | GEO: GSE218257 |
| Olfactory projection neuron scRNA-seq | Xie et al., 2021[49] | GEO: GSE161228 |
| FACS optic lobe bulk RNA-seq | Konstantinides et al., 2018[91] | GEO: GSE103772 |
| FACS central complex bulk RNA-seq | Turner-Evans et al., 2020[70] | GEO: GSE155329 |
| FACS central complex bulk RNA-seq | Wolff et al., 2025[68] | GEO: GSE271123 |
| FACS optic lobe and central complex bulk RNA-seq | Davis et al., 2020[69] | GEO: GSE116969 |

*(Continued on next page)*

*Continued*

| REAGENT or RESOURCE | SOURCE | IDENTIFIER |
|---|---|---|
| **Experimental models: Organisms/strains** | | |
| *Drosophila melanogaster*: w[*]; P{y[+t7.7] w[+mC]=10XUAS-IVS-mCD8::GFP}attP2 | Bloomington DSC | RRID:BDSC_32185 |
| *Drosophila melanogaster*: w[*]; Mi{TrojanGAL4DBD.2}VGlut[MI04979-TG4DBD.2]/CyO; MKRS/TM6B, Tb[1] | Bloomington DSC | RRID:BDSC_60313 |
| *Drosophila melanogaster*: w[*]; P{w[+mC]=UAS-Stinger}2, PBac{y[+mDint2] w[+mC]=13XLexAop2-IVS-tdTomato.nls}VK00022 | Bloomington DSC | RRID:BDSC_66680 |
| *Drosophila melanogaster*: w[*]; TI{FLP}fru[FLP]/TM3, Sb[1] | Bloomington DSC | RRID:BDSC_66870 |
| *Drosophila melanogaster*: wg[Sp-1]/CyO; Mi{Trojan-GAL4DBD.2}Gad1[MI09277-TG4DBD.2]/TM6B, Tb[1] | Bloomington DSC | RRID:BDSC_82987 |
| *Drosophila melanogaster*: TI{GAL4}dsx[GAL4]/TM6B, Tb[1] | Rideout et al. 2010[92] | N/A |
| *Drosophila melanogaster*: P{GAL4}odd[MC] / CyO | Gift from F. Casares | N/A |
| *Drosophila melanogaster*: {10x-UAS-FRT-IVS-mCD8::GFP-STOP-FRT-IVS-CsChrimson::tdTomato}attp40 | Gift from G. Miesenböck | Vrontou et al., 2021[93] |
| *Drosophila melanogaster*: w[1118]; TI{GAL4(DBD)::Zip-}dsf[GAL4(DBD)::Zip-]/CyO | Gift from T. Shirangi | Diamandi et al., 2024[94] |
| *Drosophila melanogaster*: y[1] w[*]; TI{Trojan-p65AD.1}bsh[CR00771-Tp65AD.1]/SM6a | This study | N/A |
| *Drosophila melanogaster*: y[1] w[*];; Mi{Trojan-p65AD.0}TfAP-2[MI04611- Tp65AD.0]/TM3, Sb[1] Ser[1] | This study | N/A |
| *Drosophila melanogaster*: y[1] w[*];; Mi{Trojan-p65AD.1}Ptx1[MI11305- Tp65AD.1]/TM3, Sb[1] Ser[1] | This study | N/A |
| *Drosophila melanogaster*: y[1] w[*]; Mi{Trojan-GAL4DBD.1}dve[CR70543-TG4DBD.1]/SM6a | This study | N/A |
| **Software and algorithms** | | |
| Adobe Illustrator CC | Adobe Systems, San Jose, CA | RRID:SCR_010279 |
| Alevin (salmon v1.7.0) | Srivastava et al., 2019[95] | RRID:SCR_017036 |
| alevinQC (v1.6.1) | Soneson and Srivastava, 2021[96] | N/A |
| AnnotationDbi (v1.52.0) | Pagès et al., 2020[97] | RRID:SCR_023487 |
| AUCell Bioconductor package | Aibar et al., 2017[41] | RRID:SCR_021327 |
| cgat-core (v0.6.7) | Sims et al., 2014[98] | RRID:SCR_006390 |
| clusterProfiler (v3.18.1) | Yu et al., 2012[99] | RRID:SCR_016884 |
| CMTK Registration Toolkit | https://github.com/jefferis/fiji-cmtk-gui | RRID:SCR_002234 |
| ComplexHeatmap (v2.6.2) | Gu et al., 2016[100] | RRID:SCR_017270 |
| corrr (v0.4.3) | Kuhn et al., 2020[101] | N/A |
| cowplot (v1.1.1) | Wilke, 2020[102] | RRID:SCR_018081 |
| Cytoscape | Shannon et al., 2003[103] | RRID:SCR_003032 |
| DoubletFinder (v2.0.3) | McGinnis et al., 2019[104] | RRID:SCR_018771 |
| dplyr (v1.0.5) | Wickham et al., 2021[105] | RRID:SCR_016708 |
| DropletUtils (v1.10.3) | Lun et al., 2019[106] | RRID:SCR_026136 |
| Fiji | https://fiji.sc/ | RRID:SCR_002285 |
| FlyBase | Jenkins et al., 2022[107], http://flybase.org/ | RRID:SCR_006549 |
| FlyLight | HHMI Janelia Research Campus | https://www.janelia.org/project-team/flylight |
| FlyWire | Dorkenwald et al., 2024[11]; Schlegel et al., 2024[10] | https://flywire.ai/ |
| future (v1.21.0) | Bengtsson, 2021[108] | N/A |
| ggcorrplot (v0.1.3) | Kassambara, 2019[109] | N/A |

*(Continued on next page)*

***Continued***

| REAGENT or RESOURCE | SOURCE | IDENTIFIER |
|---|---|---|
| ggplot2 (v3.3.3) | Wickham, 2016[110] | RRID:SCR_014601 |
| ggpubr (v0.4.0) | Kassambara, 2020[111] | RRID:SCR_021139 |
| harmony (v1.0) | Korsunsky et al., 2019[112] | RRID:SCR_022206 |
| Monocle3 (v1.0.0) | Cao et al., 2019[113] | N/A |
| org.Dm.eg.db (v3.12.0) | Carlson, 2020[114] | N/A |
| Patchwork (v1.1.1) | Pedersen, 2020[115] | RRID:SCR_000072 |
| pheatmap (v1.0.12) | Kolde, 2019[116] | RRID:SCR_016418 |
| pySCENIC (v0.11.2) | Van de Sande et al., 2020[42] | RRID:SCR_025802 |
| Python | Python Software Foundation | https://www.python.org |
| readr (v1.4.0) | Wickam and Hester, 2020[117] | N/A |
| R Statistical Software | R Core Team | RRID:SCR_001905 |
| RStudio | RStudio | RRID:SCR_000432 |
| Ruffus (v2.8.4) | Goodstadt, 2010[118] | RRID:SCR_022196 |
| Scrublet | Wolock et al., 2019[119] | RRID:SCR_018098 |
| Seurat (v4.1.0) | Hao et al., 2021[120] | RRID:SCR_016341 |
| SeuratObject (v4.0.4) | Satija et al., 2021[121] | N/A |
| Shiny | Chang et al., 2022[122] | RRID:SCR_001626 |
| ShinyCell | Ouyang et al., 2021[123] | RRID:SCR_022756 |
| SingleCellExperiment (v1.12.0) | Amezquita et al., 2020[124] | N/A |
| Solo (scvi-tools v0.17.1) | Bernstein et al., 2020[125] | RRID:SCR_026673 |
| SoupX | Young and Behjati, 2020[126] | RRID:SCR_019193 |
| stringr (v1.4.0) | Wickham, 2019[127] | RRID:SCR_022813 |
| SummarizedExperiment (v1.20.0) | Morgan et al., 2020[128] | N/A |
| tibble (v3.1.1) | Müller and Wickham, 2021[129] | RRID:SCR_026493 |
| tidyr (v1.1.3) | Wickham, 2021[130] | RRID:SCR_017102 |
| VVDviewer | Wan et al., 2017[131]; Lillvis et al., 2022[132] | https://github.com/JaneliaSciComp/VVDViewer |
| zoo (v1.8-9) | Zeileis and Grothendieck, 2005[133] | N/A |
| **Other** | | |
| Leica SP5 | Leica | RRID:SCR_018714 |

## EXPERIMENTAL MODEL AND STUDY PARTICIPANT DETAILS

### Drosophila stocks

All *Drosophila melanogaster* stocks were reared at 25°C and 40-50% humidity on standard cornmeal-agar food with a 12:12 light/dark cycle. Genotypes of the flies used were reported in the figure and legend. All strains used in the study are indicated in the key resources table.

### Full genotype list

Related to Figures 1, 2, S1, and S6.

| Figure | Full genotype |
|---|---|
| **1A, S1** | +/w*; *UAS-Stinger*, *13XLexAop2-IVS-tdTomato.nls*/+; *dsx^GAL4*/+ (males and females) |
| **2C, S6A-D** | *yw*/w*; VGlut^DBD*/+; *10xUAS-IVS-mCD8::GFP/Ptx1^p65AD* *yw*/w*;10xUAS-IVS-mCD8::GFP*/+; *Gad1^DBD/TfAP-2^p65AD* *yw*/w*;10xUAS-IVS-mCD8::GFP*/ *bsh^p65AD; Gad1^DBD*/+ *w*; odd^Gal4/10xUAS-IVS-mCD8::GFP* |
| **S6A,E-F** | *w*; dve^DBD*/ *UAS>mCD8::GFP>CsChrimson::tdTomato; Ptx1^p65AD/fru^FLP* (GFP and tdTomato channels merged) *w*; dsf^DBD/10xUAS-IVS-mCD8::GFP; Ptx1^p65AD*/+ |

All genotypes are female unless otherwise stated.

## METHOD DETAILS

### Central brain single-cell sample preparation

The central brain sample preparation was carried out as described previously[35]. The fly strain used *(+/w\*; UAS-Stinger, 13XLexAop2-IVS-tdTomato.nls/+; dsx^GAL4/+)* was a genetic cross between *w\*; UAS-Stinger, 13XLexAop2-IVS-tdTomato.nls* males (BDSC: 66680) and *+; +; dsx^GAL4* virgin females[92]. Flies were raised at 25°C on standard cornmeal-agar food in an incubator with a 12:12 light/dark cycle. Virgin males and females were collected and stored individually. Flies were aged 5 days post-eclosion at 25°C prior to dissection. The central brain dissociation protocol was carried out as described previously (Allen et al., 2020). 2 replicates each of 20 male and 20 female central brains were individually dissected, removing optic lobes, in toxin-supplemented ice-cold calcium- and magnesium-free DPBS (Gibco^TM, + 50 µM D(−)-2-Amino-5-phosphonovaleric acid, 20 µM 6,7-dinitroquinoxaline-2,3-dione and 0.1 µM tetrodotoxin). Each replicate was then washed in 1 mL ice-cold toxin-supplemented Schneider's medium (tSM: Gibco^TM + toxins, as above). Brains were then incubated for 30 minutes in 0.5 mL of tSM containing 1 mg/mL papain (Sigma-Aldrich) and 1 mg/mL collagenase I (Sigma-Aldrich). Brains were washed once more with tSM and subsequently triturated with flame-rounded 200 mL pipette tips. Dissociated brains were resuspended in 1 mL PBS + 0.01% BSA and filtered through a 10 µm CellTrics strainer (Sysmex, 04-0042-2314).

### Single-cell RNA-seq using 10x chromium

Libraries were made using the Chromium Single Cell 3' v2 kit from 10x Genomics. Cells were loaded in accordance with 10x Genomics documentation. The samples were sequenced with eight lanes of Illumina HiSeq4000 by the Oxford Genomics Centre. For all code and specific details of what was run, please refer to https://github.com/aaron-allen/Dmel-adult-central-brain-atlas.

### Alignment and cell identification

The fastq files were aligned to the r6.30 transcriptome release and included all transcript types. Digital expression matrices were generated using salmon-alevin tool for all the droplets per sample[95] (v1.7.0). To identify cell-containing droplets, we used the "emptyDrops" function from the DropletUtils package[106]. This method models the ambient RNA profile from low-count barcodes (<100 UMI), assumed to represent empty droplets. Each droplet is then statistically tested for significant deviation from the ambient profile using a Monte Carlo simulation. Droplets with false discovery rate (FDR)-adjusted p-values below 0.1% were classified as containing real cells. Filtered cell-containing barcodes were written out into Matrix Market (.mtx) Cellranger v2 format using the "write10xCounts" for downstream processing in Seurat. Between 300,000,000 and 500,000,000 reads per sample were processed, with approximately 68-72% of reads mapping to the transcriptome and a mean deduplication rate of 58-69%. The results from alevin were inspected with alevinQC[96] (v1.6.1). For full details, see Table S1.

### Post-alignment processing

Most downstream single-cell processing was performed in R (v4.0.2; R Core Team, 2021) with the Seurat package[120] (v4.1.0), along with many other packages including: ComplexHeatmap[100] (v2.6.2), corrr[101] (v0.4.3), cowplot[102] (v1.1.1), dplyr[105] (v1.0.5), future[108] (v1.21.0), ggcorrplot[109] (v0.1.3), ggplot2[110] (v3.3.3), ggpubr[111] (v0.4.0), Patchwork[115] (v1.1.1), pheatmap[116] (v1.0.12), readr[117] (v1.4.0), SeuratObject[121] (v4.0.4), Shiny[122], SingleCellExperiment[124] (v1.12.0), stringr[127] (v1.4.0), SummarizedExperiment[128] (v1.20.0), tibble[129] (v3.1.1), tidyr[130] (v1.1.3), zoo[133] (v1.8-9). Droplets were initially filtered to have greater than 500 UMI, greater than 300 genes, less than 15% mitochondrial UMI, and less than 15% heat-shock UMI. Doublets were predicted using three different methods: DoubletFinder[104] (v2.0.3), Scrublet[119], and Solo[125] (scvi-tools v0.17.1). Droplets were annotated as doublets if at least two of the three methods predicted them to be so (Figures S1D snd S1E). All three methods showed similar behavior in predicting doublets across distinct cell types (Figures S1E and S1F). We next estimated and removed ambient RNA contamination using SoupX[126] (v1.5.0). Ambient estimation was performed using sets of marker genes for mutually exclusive cell types (Figures S1G–S1J). The data was log-normalised using "NormalizeData", and 1000 variable genes were selected with "FindVariableFeatures" using method "mean.var.plot". The normalised counts were scaled with "ScaleData", and a principal component analysis was performed using "RunPCA". The replicates were batch-corrected and integrated with Harmony (v1.0; Korsunksy et al., 2019) using "RunHarmony". Uniform Manifold Approximation and Projections (UMAPs) and t-distributed stochastic neighbour embeddings (t-SNEs) were computed with "RunUMAP" and "RunT-SNE". Unsupervised clustering was performed by running "FindNeighbors" and "FindClusters" using the Leiden and Louvain algorithms.

### Re-processing of publicly available scRNA-seq data

Fastq files were downloaded from NCBI's Sequence Read Archive (SRA) and EMBL-EBI's European Nucleotide Archive (ENA) for the following datasets[17–27]. These datasets were all 10x Chromium 3' scRNA-seq chemistries but varied in which version (v2 and v3), as well as in tissue type (dissected whole brain, dissected central brain, dissected optic lobe, dissected retina, and whole head), and cellular preparation type (whole cell and nuclei). Although our primary focus was to achieve a central brain neuronal atlas, we included datasets of dissected and enriched optic lobe and retina samples to aid in the identification of these cell types in the central brain dissected and whole head preparations. Both Ozel et al. (2021)[22] and Kurmangaliyev et al. (2020)[23] included dissected tissue

throughout pupal development; however, we only used the pharate adult and adult time points here. All datasets were reprocessed as above to ensure consistency with our dataset and facilitate integration (Figure S2A).

### Integrating datasets

To integrate these datasets, we annotated and removed doublets but decided not to ambient correct the data. When estimating ambient contamination with SoupX, as detailed above (Figure S1), some of the datasets had predicted ambient contamination over 70%, particularly those in Park et al. (2025)[27]. Removing the estimated ambient contamination of these high ambient samples resulted in most cells being removed by UMI and gene filtering. And so, we opted not to correct for ambient contamination and instead to be vigilant in downstream analyses, ensuring that gene expression profiles were consistent across datasets. All the individual samples from each of the datasets were integrated with Harmony, using "RunHarmony" with the inclusion of dataset and preparation type (cell vs. nuclei) metadata to be corrected for, along with individual sample ID. Once integrated into a meta-head atlas (Figure S2B), non-neuronal cell types were annotated by marker gene expression and the proportion of each cluster originating from each tissue type (Figure S2C). These neuronal cells were then extracted, re-integrated, and re-clustered, generating a meta-neuronal atlas (Figure S2D). As before, peripheral and optic lobe-specific cell types were annotated by marker gene expression and cluster-wise proportion of tissue type (Figure S2E), as well as correlating to bulk-RNA-seq of known specific cell types (data not shown) and transferring the existing annotations from Janssens et al., 2022 (which was a re-analysis of the data from Davie et al., 2018) and Ozel et al., 2021 (Figure S2F). Notably, optic lobe-derived neurons lacked expression of *Imp* and *dati*, suggesting region-specific developmental programs (data not shown). With the peripheral and optic lobe neurons annotated, the remaining central brain neurons were extracted, re-integrated, and re-clustered, generating a meta-central brain neuronal atlas of 329,466 cells/nuclei (Figures 1A and S2G). A range of cluster resolutions was explored (data not shown), and a Leiden resolution of 10 was chosen for the next steps.

### Evaluating other integration methods

To evaluate the reliability of these Harmony integrations (Figure S4A), we also ran three other integration strategies to compare. We used Seurat's CCA (Figure S4B) and RPCA (Figure S4C) integrations following standard procedures, as in the following vignette - https://satijalab.org/seurat/archive/v4.3/integration_rpca. We also ran fastMNN (Figure S4D) using its SeuratWrapper[134] following standard methods, as in this vignette - https://htmlpreview.github.io/?https://github.com/satijalab/seurat.wrappers/blob/master/docs/fast_mnn.html. Clustering was performed using "FindNeighbors" and "FindClusters" in Seurat, as before for the Harmony integration. For each method, we computed the Local Inverse Simpson Index (LISI) using the "compute_lisi " function from the lisi[135] package in R, to evaluate the efficacy of the batch integration ($LISI_{(batch)}$) and the separation of cell types ($LISI_{(cell\ type)}$). All four methods resulted in similar scores of between 4.92-5.06 for $LISI_{(batch)}$ and between 1.20-1.57 for $LISI_{(cell\ type)}$. It should be noted that the fastMNN integration produced many dataset-specific clusters during unsupervised Louvain clustering, resulting in significantly lower adjusted rand indices compared to the other integration methods (Figures S4E andS4F). The similarity of the results of these established and well-performing methods (as benchmarked in[136]) gives us confidence in the validity of these results. Moving forward, we relied on the Harmony integration strategy for all other integrations and subclusterings, which yielded similar $LISI_{(batch)}$ scores (Figures S8S-S10, S12, S13, and S15–S17).

### Annotation of sex

Two of the datasets used in our meta-analysis did not separate the sexes into separate samples[17,19]. As a result, we've had to rely on in-silico sexing of these samples. We used Seurat's "AddModuleScore" function with the male-specific genes *lncRNA:roX1* and *lncRNA:roX2*. Cells with a module score less than zero were labeled female, and cells with module scores greater than zero were labeled male. To test the efficacy of this method, we performed *in silico* sexing of the datasets where the sexes were processed separately and achieved a precision of 0.93-1.00 and a recall of 0.89-0.97 for identifying a cell as female, depending on the dataset.

### Annotation of broad cell types

Broad neurotransmitter identity (cholinergic, glutamatergic, GABAergic, monoaminergic, and neuroendocrine) and special case cell types (Kenyon cells and motor neurons) were annotated using sets of marker genes. Seurat's "AddModuleScore" function was used with sets of genes specific to each broad cell type to calculate the average expression level of these gene programs across all cells. These module scores were then l2-normalised. Cells were annotated as a given broad cell type by both having a positive module score for that gene program and that l2-normalised module score being higher than any other score. The most common annotation for each cluster was then extrapolated to the remaining cells in each cluster at resolution 10. Each annotated broad cell type was then sub-clustered by extracting, re-integrating, and re-clustering only those cells (data not shown). These new sub-clustered identities were then mapped back onto the main meta-central brain neuronal atlas.

### Combinatorial cell type markers

We employed two strategies to find novel combinatorial transcription factor expression, uniquely identifying each cell type. In the first strategy, we used Seurat's "FindAllMarker" to compute all transcription factors that were significantly enriched. We then filtered them based on a between-cell-type coefficient of variation (CV) greater than 2 (Figures 1E and 1F; Table S3). Domain labels for these

transcription factors were manually reviewed and standardised to collapse synonymous or redundant domain terms - e.g., "Homeobox-like domain" and "Homeobox" were grouped under "Homeobox" (Figure 1G; Table S4). In the second strategy, we used a random forest machine-learning algorithm, with the NS-Forest package[137], to derive a minimal set of combinations of genes to classify each cell type (Table S5). These two methods generated similar, albeit differing, results. We provide both for the reader to expedite the de-orphaning of the remaining unannotated cell types.

### Gene ontology analysis

There was a total of 16,267 genes of the 16,841 total genes that had at least one UMI detected in this meta dataset. To estimate the universe or background of truly expressed genes from those arising from contamination, additional filtering was imposed. To be classified as "expressed", a gene had to have the following: (1) Summed UMI across all cells greater than or equal to 400, (2) Maximum UMI observed within a cell greater than or equal to 4, (3) The percent contribution of UMI from each dataset less than or equal to 60%, and (4) The percent contribution of UMI from the Park et al. (2025)[27] dataset less than or equal to 30%. This resulted in 9,017 "expressed" genes in the meta dataset. Significantly enriched genes in the clustered cell types were calculated with Seurat's "FindAllMarkers" using the Wilcoxon Signed-Rank test and with an average log fold change greater than 0.5 and Bonferroni-adjusted p-value less than 0.05. Gene ontology analysis was performed using the packages "clusterProfiler"[99] with "org.Dm.eg.db"[114] and "AnnotationDbi"[97]. The "enrichGO" function was used to determine the enriched terms for each of the molecular function, biological process, and cellular compartment categories (Figure S5D).

### Gene regulatory network analysis with SCENIC

pySCENIC[42] (v0.11.2) was run on select, individual subclustered clusters (Figures 2C and 6) to identify transcription factor (TF) based regulons by using the gene expression data as input. Briefly, the raw expression matrix was filtered to retain genes expressed in >1% of cells and with a count $>3 \times 0.01 \times$ number of cells. Modules comprising transcription factors and co-expressed genes were generated using GRNBoost2, then pruned to remove indirect targets lacking enrichment for the corresponding transcription factor motif and further refined (cisTarget). To build the final set of TF regulons, the predicted target genes of each TF module that show enrichment of any motif of the given TF are then merged. Due to stochasticity in gene regulatory network inference using GRNBoost2, each pySCENIC run can identify a different number of regulons and different target genes for each transcription factor. Thus, pySCENIC was run 100 times. High-confidence regulons were defined as those that occurred in >50% of runs and contained at least five high-confidence target genes. High-confidence target genes were those found within a regulon in >50% of runs. Cells were then scored for the activity of each high-confidence regulon (including only high-confidence target genes) using the AUCell Bioconductor package[41]. Briefly, single cells are scored by calculating the enrichment of a regulon, which is measured as the Area Under the recovery Curve (AUC) across the ranking of all genes in a particular cell, whereby genes are ranked by their expression values. AUC represents the proportion of expressed genes in the signature and their relative expression values compared to the other genes within the cell. The output of this step is a matrix with the AUC score for each gene set in each cell. We used either the AUC scores (across regulons) directly as continuous values to cluster single cells or a binary matrix generated using a cutoff of the AUC score for each regulon. These steps were run as a pipeline written in the cgat-core ruffus framework[98,118]. As described above, AUC scores per cell were fed into Seurat and processed as a separate assay. Cell-type enriched regulons were calculated using Seurat's "FindAllMarkers" function. Gene regulatory networks were visualized with Cytoscape[103]. For visualization purposes, GRNs were further filtered to only include positive regulons less than 100 genes in size, and each node (gene) had to have a maximum expression of at least 3 UMI and total expression of at least 80 UMI.

### Early- vs. late-born morphology quantification

To annotate early-born ("*Imp>dati*") and late-born ("*Imp<dati*"), we used Seurat's "AddModuleScore" to calculate the average expression level of early (*Imp*, *mamo*) and late (*dati*, *pros*) gene programs across all cells (Figures 3C, S11A, and S11B). A cell was annotated as early-born if the l2-normalised early-born module score was both greater than zero and greater than the l2-normalised late-born module score. Conversely, a cell was annotated as late-born if the l2-normalised late-born module score was both greater than zero and greater than the l2-normalised early-born module score. We used three separate methods to quantify the "punctate" vs "serpentine" t-SNE morphology differences between early-born and late-born neurons: (1) Mean pairwise distance, (2) Fragmentation score, (3) Modularity score (Figures 3G, 3H, and S11C–S11G). Before calculating these metrics, Kenyon cells and cells where the early- and late-born module scores were either both negative or equal were removed. The mean pairwise distance in t-SNE space between all cells within each cluster was calculated with the "dist" function (Figure 3G). Cells were then split into early-born ("*Imp>dati*") and late-born ("*Imp<dati*"), as described above. Secondly, the fragmentation score is the number of subclusters generated using the "dbscan" function with an epsilon neighbourhood radius ("eps") of 0.5 and the number of neighbourhood members ("minPts") equal to 5 (Figure 3H). A range of values for "eps" and "minPts" were tested, and they all generated a difference (Figures S11D and S11E). Thirdly, we calculated a modularity score for each cell in a cluster-wise fashion. Distances between points in t-SNE space were calculated with the "dist" function. Graphs were calculated with "graph_from_adjacency_matrix" and communities were detected with "cluster_fast_greedy". Lastly, the modularity of these communities was calculated with the "modularity" function (Figure S11G). The differences were statistically quantified using the Wilcoxon Signed-Rank test with Bonferroni correction.

### Annotating and subclustering ALad1

The ALad1 olfactory projection neuron lineage is co-positive for *acj6* and *Oaz*[25]. We extracted the cells from cell type "Achl 032", the only cluster co-positive for these genes, as an initial proxy. These data were integrated, as above, with previously published scRNA-seq data from the developing olfactory projection neurons from multiple hemilineages[49] that were downloaded from the Gene Expression Omnibus (GSE161228). In this integration of datasets, the hemilineage identity, annotated in Xie et al. (2021)[49], was extrapolated to the cells from the other datasets by cluster. Clusters from the meta-central brain neuron atlas now annotated as ALad1 (*acj6*+, *Oaz*+, and *vvl*-) were extracted, re-integrated, and re-clustered, as above, to form our refined proxy for the adult ALad1 hemilineage (Figure S12).

### Pseudotime analysis

Pseudotime analysis was performed using Monocle3[113] (v1.0.0) using standard methods. Briefly, the data were processed with functions "new_cell_data_set", "preprocess_cds", "align_cds", "reduce_dimension", "cluster_cells", and "learn_graph" using default settings. Cells were ordered with "order_cells" setting all *Imp*+ cells as the root. Genes with significantly variable expression across pseudotime were calculated with "graph_test" and filtered for q-value less than 0.05 and Moran's I greater than 0.1. Heatmaps of significantly varying genes were plotted with the package "ComplexHeatmap". For the systematic pseudotime analysis across the central brain cell types (Figures 4H–4J), only clusters at Leiden resolution 10 that met the following criteria were used: (1) predicted type I or type II lineages, (2) l2-normalised late-born module score (see above) greater than zero and greater than l2-normalised early-born module score, (3) a single sub-region of the cluster positive for *br*. A total of 75 of the 220 clusters met these criteria and were subclustered and subjected to pseudotime analyses. Following the standard processing described above, the resulting pseudotime metadata for each of these clusters was normalised to a percent to be comparable to each other. To identify genes that repeatedly have temporal transcription factors that vary over pseudotime, we filtered the resulting pseudotime markers for each of the clusters for transcription factors that varied significantly (q-value < 0.05, Moran's I > 0.1) in at least 50% clusters (Figure 4J). Domain labels for these transcription factors were manually reviewed as above (Figure 4K; Table S6).

### Annotating and subclustering type II NSC lineages

To annotate neurons derived from type II lineages in our central brain neuronal atlas, we leveraged previously published scRNA-seq of FAC-sorted late L3 larval type II lineages[65,66]. Fastq files from these datasets were downloaded and re-processed as described above. After an initial integration, the central brain neuronal population of these data were extracted, re-integrated, and re-clustered to form a larval type II neuronal atlas. These larval data were then integrated with a subset of our reprocessed adult datasets. For this integration, we selected only whole-cell preparation datasets with the Kenyon cells removed. Resulting clusters at resolutions of 0.2, 1, 2, 4, and 10 that have more than 5% of their cells derived from the larval type II datasets were annotated as "type II", and all others were annotated as "type I" (Figure 5B). These lineage-type annotations were then extrapolated, in a cluster-wise fashion, to the rest of the meta-central brain neuron atlas (Figure 5C). Once annotated, these type II neurons were extracted, re-integrated, and re-clustered, as above, to generate an adult type II neuronal stem cell lineage atlas (Figure 5D). Type II cell types were annotated by a combination of correlation to bulk-RNA-seq data (as above, Figure S13), marker gene expression (Figure S14), and genetic intersection[29] (Figure 2).

### Annotating and subclustering neuroendocrine and monoaminergic neurons

Neuroendocrine clusters were annotated as clusters with a positive z-scored expression of the transcription factor *dimm*. These cells were extracted, re-integrated, and re-clustered as described above. After this initial subclustering, a few cell types were negative for *dimm* as they were part of a heterogeneous parental cluster. These cells were removed, and the remaining cells were re-subclustered. Gene regulatory network analysis was performed on the neuroendocrine cell types as described above. Subclustering of the monoaminergic neurons was conducted in a similar fashion (Figure S16). Broad monoaminergic class was assigned by the following marker genes: dopaminergic (*ple*+ and *DAT*+), histaminergic (*Hdc*+), octopaminergic (*Tdc2*+ and *Tbh*+), serotonergic (*SerT*+ and *Trh*+), and tyraminergic (*Tdc2*+ and *Tbh*-). Further subclustering of the non-PAM (Figures S17A–S17C) and PAM (Figures S17D–S17G) subdivisions of monoaminergic clustering was performed by separating the non-PAM cells from the PAM (*DAT*+, *Fer2*+, and *Imp*-) cells.

### Generating a web app

Our interactive web tool for visualization of these scRNA-seq data was generated using a modified version of "ShinyCell", an R-based Shiny App[123]; https://github.com/SGDDNB/ShinyCell). Specifically, we forked and modified "easyshiny" (https://github.com/NBI Sweden/easyshiny), which is a forked version of the original "ShinyCell". Our modified version is available here - https://github.com/aaron-allen/easyShinyCell.

### Generation of split-Gal4 driver lines

We used existing coding intronic Minos-mediated integration cassette/CRISPR-mediated integration cassette (MiMIC/CRIMIC) lines[138,139] (see key resources table) to generate split-GAL4 drivers using the Trojan method[140]. pBS-KS-attB2-SA()-T2A-Gal4DBD-Hsp70 or pBS-KS-attB2-SA()-T2A-p65AD-Hsp70 vectors[140] with the appropriate reading frame were inserted in the MiMIC/CRIMIC

locus of a given line via recombinase-mediated-cassette-exchange through injection (BestGene, CA). Stocks were generated with the transformed flies as described in[140].

### Immunohistochemistry

After a brief pre-wash of adult flies in 100% EtOH to remove hydrophobic cuticular chemical compounds, brains were dissected in PBS at RT (20-25°C), collected in 2 mL sample tubes and fixed with 4% formaldehyde (Sigma-Aldrich) in PBS (Sigma-Aldrich) for 20 min at RT. After fixation, tissues were washed in 0.5-0.7% Triton X-100 in PBS (Sigma-Aldrich) (PBT) 3 times each for 20 min at RT. After blocking in 10% normal goat serum (Sigma-Aldrich) in PBT (NGS/PBT) overnight (8-12 h) at RT, tissues were incubated in primary antibody solutions for 48-72 hrs at 4°C (1:1000, rabbit anti-GFP, Thermo Fisher Scientific; 1:1000, chicken anti-GFP, Abcam; 1:1000, rabbit anti-RFP, antibodies-online; 1:10, mouse anti-Brp, Developmental Studies Hybridoma Bank). After four washes in PBT for 1 hr each at RT, tissues were incubated in secondary antibody solutions for 48 hours at 4°C (1:500, anti-rabbit Alexa Fluor 488, anti-chicken Alexa Fluor 488, anti-mouse Alexa Fluor 546, anti-rabbit Alexa Fluor 546, anti-mouse Alexa Fluor 633, Thermo Fisher Scientific). After four washes in PBT for 1 hr each at RT, specimens were imaged directly, or 70% glycerol in PBS was added to the sample tubes, which were subsequently transferred to -20°C and kept for at least 8 hr for tissue clearing. Specimens were mounted in Vectashield (Vector Laboratories).

### Confocal image acquisition and processing

Confocal image stacks were acquired on a Leica TCS SP5 confocal microscope at 1024 x 1024-pixel resolution with a slice size of 0.29 μm or 1 μm. Water-immersion 25x and oil-immersion 40x objective lenses were used for brain images. Images were registered onto the intersex template brain using the Fiji Computational Morphometry Toolkit (CMTK) Registration GUI (https://github.com/jefferis/fiji-cmtk-gui). For segmented images, we used the software VVDViewer[131,132] (https://github.com/JaneliaSciComp/VVD Viewer) to render the registered image stacks in 3D, manually mask other neurons co-labeled in the image and segment out neurons of interest. All segmented images include unsegmented versions within the supplemental figures.

