## [Document S2. Transparent peer review records for Allen et al. · Cell Genomics]

A High-Resolution Atlas of the Brain Predicts Lineage and Birth Order Underlie Neuronal Identity

Aaron M. Allen, Megan C. Neville, Tetsuya Nojima, Faredin Alejevski, Devika Agarwal, David Sims, and Stephen F. Goodwin

Summary

Initial submission: Received : Jun 27, 2025

Scientific editor: Sara Rohban

First round of review: Number of reviewers: 2
Revision invited : Jul 31, 2025
Revision received : Oct 20, 2025

Second round of review: Number of reviewers: 1
Accepted : Nov 15, 2025

Data freely available: YES

Code freely available: YES

This transparent peer review record is not systematically proofread, type-set, or edited. Special characters, formatting, and equations may fail to render properly. Standard procedural text within the editor's letters has been deleted for the sake of brevity, but all official correspondence specific to the manuscript has been preserved.

Referees' reports, first round of review

Reviewer #1:

It is a beautiful study, taking advantage of the previously published datasets and integrating them, authors generated fly brain scRNAseq atlas at an unprecedented detail. one can identify needle in a haystack using this study. I support the acceptance of the paper as is but I also suggest the following minor modifications:

Line 130-133: it is a little bit cryptic. Author should clearly and directly state their integrated data resolved this complexity.

Line 165-166: entire might be an overstatement for some lineages. e.g., for DM2, EM appears to mark more cells than the gene intersection does.

Line 176-178: It would be nice to explain SCENIC a little bit more for the reader outside of the field.

Line 284: I would classify MB NB as type I. I would check the literature to make sure MB NBs can be classified as type 3.

Line 325-327: Do authors think these 34 subcluster might represent distinct developmental origin? If yes, I would mention it.

Line 329-330: Citing references for the distinct origins needed.

Line 353: A reference needed Otp binding Hug.

Line 576-581: it is not clear what 10 subclasses are in the graphs of Fig 4E, G. Why 10 is chosen?

Reviewer #2:

Allen et al. present a compelling and comprehensive single-cell transcriptomic examination of the central brain of *Drosophila melanogaster*. Producing a highly-resolved molecular map of the central brain in this model organism is of considerable value for the research community. Based on their analysis, the authors propose a model in which (1) neuroblast hemilineages are the primary determinant of the transcriptomic cell type identity in the mature brains, and (2) the birth order of neurons within each hemilineage leaves a molecular signature that is common across hemilineages and maintained into adulthood. The conclusions are interesting and important, and the integration of multiple datasets (single-cell and single-nuclei, multiple different labs) with increased sequencing coverage is a powerful approach. However, the authors should provide stronger evidence that the integration process did not introduce technical artifacts that might have influenced the study's overarching conclusions. Moreover, a deeper examination of the transcription factors orchestrating the GRNs highlighted in the study is needed, as well as a practical roadmap for the deorphanization of the proposed genetically-defined hemilineages.

Major comments:

1. The authors' effort to integrate multiple datasets and increase the sequencing coverage is a powerful approach. They also present some analyses (for example Figure S3panel B) that indicate all datasets are well represented across the 246 "broad" cellular identities represented in the major distinct neuronal clusters, which seem to map onto entire hemilineages. However, many of the paper's key claims are based not on these broad clusters but on their further sub-clustering into "subtypes" within each hemilineage. Examples include, Fig. 2D-E-F, Fig. 3, 4, Figure S6, statements such as "we can resolve 4,167 distinct neuronal subtypes offering an unprecedented view of adult brain cellular diversity", and the paragraph in lines 184~196.

Therefore, it is critical that the authors more rigorously demonstrate that the observed transcriptional heterogeneity of subclusters does not arise from batch effects or technical differences between datasets.

Currently, the analysis relies only on one integration approach (Harmony) although many are available (reviewed for example in Heumos et al 2023, with additional recent methods since then). The authors should apply and compare multiple algorithms and provide robustness and benchmarking metrics. Such comparisons could reveal low-quality integrations within broad clusters, potentially requiring the removal of artifactual subclusters that reflect dataset-specific biases rather than genuine transcriptional identities.

The authors should provide visual projections (UMAP/tSNE) of subclusters color-coded by dataset-of-origin, and metrics of cross-dataset representations for biologically relevant contexts (how are the datasets-of-origin represented in all subclusters that are Imp+, in all subclusters that are dati+, in all neuroendocrine subclusters, in all aminergic subclusters, etc.). The goal should be to demonstrate that all the subclusters in the final atlas are consistently represented in multiple datasets-of-origin and are not artifacts of batch correction or dimensionality reduction.

2. Line 179 - "We can resolve 4,167 distinct neuronal subtypes offering an unprecedented view of adult brain cellular diversity". The authors should clarify that this number is an upper bound rather than a precise or best estimate of neuronal subtype diversity. Dimensionality reduction and embedding methods are known to overemphasize subtle expression differences that do not necessarily correspond to biologically unified and distinct cell types or subtypes (see ongoing debate about cell type versus cell state, but also simply biologically-irrelevant noise in data). The computational delimitation of a UMAP subcluster after multiple rounds of clustering is not a sufficient criterion to call for a bona fide distinct neuron subtype.

3. The notion of hierarchy in the gene regulatory networks and sets of transcription factors determining clusters and cell types is lacking. For example, do TFs that determine broad identity (hemilineages) and TFs that determine subtypes (correlative with birth order) belong to different or to the same TF families (Homeodomain, bHLH, Zinc finger, etc)? Are there TFs that act both as vertical hemilineage specifiers and as horizontal subtype specifiers across distinct hemilineages, or are these sets mutually exclusive? This would be important in order to anchor the conceptual model proposed to concrete genes amenable for genetic experiments, and to place the model in relationship to the evolution of TF families and evolution of cell types (see for example reviews by Detlev Arendt and colleagues). The authors should provide supplementary tables that state:

- for each proposed lineage, the putative combination of hemilineage-specifying TFs.
- A list of the TFs participating in the putative "repeated temporal transcription factor programs", including their placement along the early-late axis, and metrics of how frequently each TF correlates with temporal identity (e.g. , in how many identities out of the 246 broad identities is TF X marking late-

born neurons).

4. Related to the above, while it is unrealistic to expect the authors to apply the intersectional genetic strategy to "solve" all the hemilineages in the present study, the authors are in a great position to provide a list of the minimal sets of genes that could be used to genetically trace every cluster/hemilineage and distinguish it from others. In the same way that Ptx1 \cap VGlut was selected and experimentally applied for one cluster, but for all clusters. This would be a tremendous resource for the community and an important step towards matching the transcriptional and anatomical maps. Moreover, it would provide a concrete and testable set of predictions that could corroborate or challenge the study's central claims in future work.

Minor comments

5. Sets of TFs that define an entire "broad" cluster/hemilineage should be displayed also in the heatmap discussing the subclusters wherein. For example, Figure 2D should display the heatmap for Ptx1 expression (which is expected to be expressed in all the subclusters, according to the authors). And if other TFs display the same expression pattern as Ptx1 (hemilineage-specific AND pan-intra-hemilineage, like Ptx1), they should appear together with it as well. This is important both for clarity of presentation and because of the biological significance. There is a hierarchical relationship between the broad(/hemilineage)-defining transcription factors (Ptx1) and the sub-clusters within it, that are responsible for the heterogeneity of cell types within a single hemilineage. Same goes for significantly-enriched regulons - those regulons that are "pan-enriched" in the entire "broad cluster" across all its subclusters should be represented in the heatmap, so that the differential sub-regulons will appear in context of the commonly-shared regulons.

6. Please be more explicit about cases where transcriptionally-defined cell types only approximately represent hemilineages (eg. CREa1, lines 164-166).

Authors' response to the first round of review

RESPONSE TO REVIEWERS

Reviewer #1: It is a beautiful study, taking advantage of the previously published datasets and integrating them, authors generated fly brain scRNAseq atlas at an unprecedented detail. one can identify needle in a haystack using this study. I support the acceptance of the paper as is, but I also suggest the following minor modifications:

Line 130-133: it is a little bit cryptic. Author should clearly and directly state their integrated data resolved this complexity.

We thank the reviewer for this suggestion. We have extended our statement in Line 135-138: “Indeed, no individual dataset used here was sufficient on its own to describe this complexity, instead our cost-effective approach of integrating multiple datasets achieved the necessary resolution to identify and annotate the vast majority of central brain neurons.”

Line 165-166: entire might be an overstatement for some lineages. e.g., for DM2, EM appears to mark more cells than the gene intersection does.

We acknowledge that light microscopy (LM) images in Figure 2C may appear less complete than EM images. In cases like DM2, weaker posterior projections can appear faint or absent in LM due to signal attenuation with imaging depth, whereas EM preserves consistent contrast across the brain. We maintain that the entire hemilineage is present in our LM images and have replaced the DM2 image to better reflect its expression pattern. In addition, we have now indicated that all corresponding LM data will be deposited in Virtual Fly Brain (<https://www.virtualflybrain.org/>), enabling 3D inspection.

Line 176-178: It would be nice to explain SCENIC a little bit more for the reader outside of the field.

We have now added some details to Line 194-200: “To investigate the regulatory logic underlying this diversity, we focused on the SMPad1 hemilineage (Figure 2D-K, Figure S8). Subclustering analysis revealed 22 transcriptionally distinct subtypes, with hemilineage- and subtype-specific transcription factor expression (Figure 2D). We used SCENIC^{42,43} to infer gene regulatory networks (GRNs), integrating gene co-expression with DNA motif enrichment to predict transcription factors and their putative target genes (regulons; Figure 2E).”

Line 284: I would classify MB NB as type I. I would check the literature to make sure MB NBs can be classified as type 3.

Thank you for highlighting this, although there are inconsistencies in the literature, contemporary publications have been referring to MB NBs as type I, thus we have removed it as a NB type.

Line 325-327: Do authors think these 34 subcluster might represent distinct developmental origin? If yes, I would mention it.

Yes, we believe they do represent distinct developmental origins. This is addressed in Line 388-390: “Despite their shared functional role in neuropeptide release, neuroendocrine cell types arise from different hemilineages, giving each subtype distinct transcriptional signatures.”

Line 329-330: Citing references for the distinct origins needed.

Done

Line 353: A reference needed Otp binding Hug.

We realise this statement was ambiguous and it has been adjusted in Line 405-408 for clarity: “We focused on the homeodomain transcription factor Otp and its regulon, as SCENIC analysis predicts Otp regulates Hug, and its regulon showed restricted expression across neuroendocrine subtypes, particularly in three Hug-expressing subtypes (Figure 6FG).”

Line 576-581: it is not clear what 10 subclasses are in the graphs of Fig 4E, G. Why 10 is chosen?

We thank the reviewer for raising this question. We had selected 10 k-means clusters to provide a balance between interpretability and resolution of gene expression dynamics across pseudotime. Since the exact number of clusters is somewhat subjective, we have changed the clustering of rows to hierarchical clustering using the Ward's method (ward.D2 specifically), thereby removing any subjectivity in selecting the number of clusters, as in kmeans.

Reviewer #2: Allen et al. present a compelling and comprehensive single-cell transcriptomic examination of the central brain of *Drosophila melanogaster*. Producing a highly-resolved molecular map of the central brain in this model organism is of considerable value for the research community. Based on their analysis, the authors propose a model in which (1) neuroblast hemilineages are the primary determinant of the transcriptomic cell type identity in the mature brains, and (2) the birth order of neurons within each hemilineage leaves a molecular signature that is common across hemilineages and maintained into adulthood. The conclusions are interesting and important, and the integration of multiple datasets (single-cell and single-nuclei, multiple different labs) with increased sequencing coverage is a powerful approach. However, the authors should provide stronger evidence that the integration process did not introduce technical artifacts that might have influenced the study's overarching conclusions. Moreover, a deeper examination of the transcription factors orchestrating the GRNs highlighted in the study is needed, as well as a practical roadmap for the deorphanization of the proposed genetically-defined hemilineages.

We thank the reviewer for their thoughtful and constructive comments on our manuscript. We are pleased that they recognised the value of our atlas to the field, as well as the strength of our integrative approach and the model linking transcriptomic identity to neuronal hemilineage and birth order. In response to Reviewer #2's main concerns, we have made several revisions to further strengthen the manuscript. A detailed description of these updates is provided below. We believe these additions clarify and enhance the significance and impact of our findings.

Major comments:

1. The authors' effort to integrate multiple datasets and increase the sequencing coverage is a powerful approach. They also present some analyses (for example Figure S3panel B) that

indicate all datasets are well represented across the 246 "broad" cellular identities represented in the major distinct neuronal clusters, which seem to map onto entire hemilineages. However, many of the paper's key claims are based not on these broad clusters but on their further sub-clustering into "subtypes" within each hemilineage. Examples include, Fig. 2D-E-F, Fig. 3, 4, Figure S6, statements such as "we can resolve 4,167 distinct neuronal subtypes offering an unprecedented view of adult brain cellular diversity", and the paragraph in lines 184~196.

Therefore, it is critical that the authors more rigorously demonstrate that the observed transcriptional heterogeneity of subclusters does not arise from batch effects or technical differences between datasets.

Currently, the analysis relies only on one integration approach (Harmony) although many are available (reviewed for example in Heumos et al 2023, with additional recent methods since then). The authors should apply and compare multiple algorithms and provide robustness and benchmarking metrics. Such comparisons could reveal low-quality integrations within broad clusters, potentially requiring the removal of artifactual subclusters that reflect datasetspecific biases rather than genuine transcriptional identities.

The authors should provide visual projections (UMAP/tSNE) of subclusters color-coded by dataset-of-origin, and metrics of cross-dataset representations for biologically relevant contexts (how are the datasets-of-origin represented in all subclusters that are Imp+, in all subclusters that are dati+, in all neuroendocrine subclusters, in all aminergic subclusters, etc.). The goal should be to demonstrate that all the subclusters in the final atlas are consistently represented in multiple datasets-of-origin and are not artifacts of batch correction or dimensionality reduction.

We thank the reviewer for raising these important points. To address concerns about potential artifacts introduced by dataset integration, we have performed additional analyses using multiple alternative integration methods. We now compare four commonly used integration methods: Harmony (as used in the original manuscript), Seurat v4 CCA, Seurat v4 RPCA, and fastMNN and have included metrics for cross-data representation (Figure S4). All methods produced consistent high-level integration and showed strong agreement in cluster assignment, apart from fastMNN which generated many small dataset-specific clusters. Thus, clusters identified using Harmony were robustly recovered by other methods, reinforcing their biological validity.

As suggested, we have now included visual projections (UMAP/tSNE) of the central brain atlas and all biologically relevant subclusters color-coded by dataset-of-origin and have included LISI (Local Inverse Simpson's Index) scores (Figures S3, S4, S7, S9, S10, S12-13, and S15-17), confirming that they are not artifacts of integration or batch correction.

Together, these results demonstrate that our conclusions are robust to the choice of integration method. We have provided a description of all further analysis in the methods: "Evaluating other integration methods".

2. Line 179 - "We can resolve 4,167 distinct neuronal subtypes offering an unprecedented view of adult brain cellular diversity". The authors should clarify that this number is an upper bound rather than a precise or best estimate of neuronal subtype diversity. Dimensionality reduction and embedding methods are known to overemphasize subtle expression differences that do not necessarily correspond to biologically unified and distinct cell types or subtypes (see ongoing debate about cell type versus cell state, but also simply biologically irrelevant noise in data). The computational delimitation of a UMAP subcluster after multiple rounds of clustering is not a sufficient criterion to call for a bona fide distinct neuron subtype.

We appreciate the reviewer's concern and agree that high-resolution clustering can capture subtle differences that may not correspond to stable or functionally distinct cell types. The reported 4,167 subtypes reflect the granularity of our clustering approach and not a fixed or exhaustive representation of neuronal identities. However, we include multiple independent validations, via genetic intersection, of subclustered cell types in our companion paper, which we now explicitly reference in the Discussion (line 508). We also have another manuscript in preparation exploring one of these subclustered hemilineages, where we rigorously validate all transcriptionally defined subtypes, linking them to their established anatomical subtypes (McGinnis et al., in prep). Furthermore, estimates based on connectome-derived anatomy indicate that there are 7,396 central brain neuronal cell types (Schlegel et al., 2024), suggesting that our current estimate of 4,167 may in fact be an underestimate of neuronal diversity.

We contend that the subclusters we identify are, for the most part, not computational artifacts. We have included new plots (Figures S3, S4, S7, S9, S10, S12-13, and S15-17) showing that these subclusters are robustly and consistently represented across all independent datasets, despite substantial biological and technical heterogeneity. This crossdataset reproducibility argues strongly against the idea that the observed diversity is driven by batch effects or integration artifacts.

We also do not believe that these distinct subclusters reflect different cell states within an otherwise uniform cell type. Given that the datasets encompass a wide range of statedependent biological perturbations—including aging (Davie et al., 2018; Lu et al., 2023), cocaine exposure (Baker et al., 2020), sleep deprivation (Dopp et al., 2024), water deprivation (Park et al., 2022), and an Alzheimer's disease model (Park et al., 2025)—it seems unlikely that all datasets would converge on the same set of transient states. Moreover, because the datasets are so effectively integrated, even at this high level of subclustering resolution, we interpret these subclusters as bona fide cell types rather than state-specific variants.

We recognise that this discussion raises a deeper conceptual question: what defines a "cell type," and at what point does a quantitative difference become a qualitative one? In our view, the answer depends on the specific context and level of resolution—developmental, anatomical, molecular, or functional—at which the term is applied, and thus its usage should remain flexible. Based on the reproducibility of our clustering across diverse datasets and its strong correspondence with known biological identities, we propose that our subclusters represent genuine and biologically meaningful transcriptional identities, and therefore bona fide cell types. Moreover, given the fine-grained anatomical definitions of a cell types described in the recent connectomes, we now emphasise in the Discussion that transcriptional and anatomical axes represent complementary and equally complex dimensions of neuronal identity (lines 556-558).

3. The notion of hierarchy in the gene regulatory networks and sets of transcription factors determining clusters and cell types is lacking. For example, do TFs that determine broad identity (hemilineages) and TFs that determine subtypes (correlative with birth order) belong to different or to the same TF families (Homeodomain, bHLH, Zinc finger, etc)? Are there TFs that act both as vertical hemilineage specifiers and as horizontal subtype specifiers across distinct hemilineages, or are these sets mutually exclusive? This would be important in order to anchor the conceptual model proposed to concrete genes amenable for genetic experiments, and to place the model in relationship to the evolution of TF families and evolution of cell types (see for example reviews by Detlev Arendt and colleagues). The authors should provide supplementary tables that state:

- for each proposed lineage, the putative combination of hemilineage-specifying TFs. - A list of the TFs participating in the putative "repeated temporal transcription factor programs", including their placement along the early-late axis, and metrics of how frequently each TF correlates with temporal identity (e.g. , in how many identities out of the 246 broad identities is TF X marking late-born neurons).

Thank you for these beneficial comments. We have added additional analyses that we believe will be of broad interest to the community. Specifically, we now compare the transcription factor families associated with hemilineage identity and those linked to birth order (Figures 1G and 4K). This analysis supports a hierarchical model of transcriptional regulation, in which distinct sets of transcription factors act along spatial and temporal axes to specify neuronal identity (Lines 308-328). In addition, we have included a supplementary table listing the combinations of hemilineage-defining transcription factors across the central brain (Table S3 and Table S4), as well as a table of temporal transcription factors (Table S5) that references Figure 4J, summarising their placement along the early-late axis and their frequency across hemilineages (Figure 4J).

Moreover, two independent studies have reported highly convergent sets of recurrent temporal factors underlying the transcriptional identities of multiple central brain lineages (Elkahlah et al., 2025) and all VNC lineages (Cachero et al., 2025) during mid-pupal development. This convergence provides strong corroborative evidence for our conclusions and further indicates that these transcriptional programs are stably maintained into adulthood.

4. Related to the above, while it is unrealistic to expect the authors to apply the intersectional genetic strategy to "solve" all the hemilineages in the present study, the authors are in a great position to provide a list of the minimal sets of genes that could be used to genetically trace every cluster/hemilineage and distinguish it from others. In the same way that Ptx1 \cap VGlut was selected and experimentally applied for one cluster, but for all clusters. This would be a tremendous resource for the community and an important step towards matching the transcriptional and anatomical maps. Moreover, it would provide a concrete and testable set of predictions that could corroborate or challenge the study's central claims in future work.

Thank you for this suggestion. We agree that systematically defining genetic access strategies adds great value to our resource. To facilitate this, we have added a schematic (Figure 7B) that outlines the logic for identifying genetic addresses to target neuronal cell types in the adult central brain. In addition to Table S2, which provides markers for transcriptionally defined clusters, we have included Table S3 (Seurat markers filtered for high between-cluster coefficient of variation) and Table S4 (NSForest

combinatorial markers), which list the combinations of cell type-defining transcription factors that could serve as minimal intersectional gene sets across the central brain.

Minor comments

5. Sets of TFs that define an entire "broad" cluster/hemilineage should be displayed also in the heatmap discussing the subclusters wherein. For example, Figure 2D should display the heatmap for Ptx1 expression (which is expected to be expressed in all the subclusters, according to the authors). And if other TFs display the same expression pattern as Ptx1 (hemilineage-specific AND pan-intra-hemilineage, like Ptx1), they should appear together with it. This important both for clarity of presentation and because of the biological significance. There is a hierarchical relationship between the broad(/hemilineage)-defining transcription factors (Ptx1) and the sub-clusters within it, that are responsible for the heterogeneity of cell types within a single hemilineage. Same goes for significantly-enriched regulons - those regulons that are "pan-enriched" in the entire "broad cluster" across all its subclusters should be represented in the heatmap, so that the differential sub-regulons will appear in context of the commonly-shared regulons.

We thank the reviewer for this helpful suggestion. We have now included hemilineage-wide TFs and regulons to our analysis in Figure 2D and E.

6. Please be more explicit about cases where transcriptionally-defined cell types only approximately represent hemilineages (e.g., CREa1, lines 164-166).

We had included approximations in Figure 2 due to discrepancies between our definition of cells within a hemilineage and those currently used to assign a hemilineage in Codex, which largely excludes embryonic born neurons. We have removed the approximation symbols in Figure 2 and have adapted the line in the text: "Using this approach, we repeatedly and consistently (8/8) found that intersected transcriptionally defined cell types represent hemilineages^{29,40} (Figure 2C; Figure S6), just as we and others had previously seen in the VNC^{35,41}" (lines 182-185).

Referees' reports, second round of review

Reviewer #2: The authors have done an excellent job in addressing the reviewer's comments.